# Regulation of subcellular dendritic synapse specificity by axon guidance cues

**Emily C Sales[1,2], Emily L Heckman[1,2], Timothy L Warren[1,2], Chris Q Doe[1,2]***

[1]Institute of Neuroscience, Howard Hughes Medical Institute, University of Oregon, Eugene, United States; [2]Institute of Molecular Biology, Howard Hughes Medical Institute, University of Oregon, Eugene, United States

**Abstract** Neural circuit assembly occurs with subcellular precision, yet the mechanisms underlying this precision remain largely unknown. Subcellular synaptic specificity could be achieved by molecularly distinct subcellular domains that locally regulate synapse formation, or by axon guidance cues restricting access to one of several acceptable targets. We address these models using two *Drosophila* neurons: the dbd sensory neuron and the A08a interneuron. In wild-type larvae, dbd synapses with the A08a medial dendrite but not the A08a lateral dendrite. dbd-specific overexpression of the guidance receptors Unc-5 or Robo-2 results in lateralization of the dbd axon, which forms anatomical and functional monosynaptic connections with the A08a lateral dendrite. We conclude that axon guidance cues, not molecularly distinct dendritic arbors, are a major determinant of dbd-A08a subcellular synapse specificity.

DOI: https://doi.org/10.7554/eLife.43478.001

## Introduction

Nervous system function is determined by the precise connectivity of neurons. From the *Drosophila* larva with 10,000 neurons to the human with 80 billion neurons, all neurons are faced with the challenge of identifying the correct subset of synaptic partners among many potential target neurons. In addition to specificity at a cellular level, neural circuits also exhibit synaptic specificity at the subcellular level (reviewed in *Yogev and Shen, 2014*). In *Drosophila*, the giant fiber descending neuron targets a specific dendritic domain of the tergotrochanteral motor neuron in a fast jump escape circuit (*Godenschwege et al., 2002*; *Godenschwege and Murphey, 2009*). In mammals, cortical pyramidal neurons receive input from martinotti neurons on their distal dendrites and basket neurons on their proximal dendrites (*Huang et al., 2007*) (*Figure 1A*). The precise targeting of inhibitory neurons to distinct subcellular domains of their target neurons has profound effects on neural processing and circuit function by gating action potential initiation, providing a substrate for plasticity, altering mEPSP amplitude, and modulating dendritic integration (*Bloss et al., 2016*; *Hao et al., 2009*; *Miles et al., 1996*; *Pouille et al., 2013*; *Tobin et al., 2017*). Although the precise subcellular positioning of synapses is important for proper circuit function, the mechanisms necessary to achieve such specificity are just starting to be explored (*Telley et al., 2016*).

Two distinct developmental models could explain subcellular synaptic specificity. The first model relies on molecular differences between two subcellular domains to restrict synapse formation to one domain (the 'labeled arbor' model). This model is supported by evidence in mouse and *C. elegans* whereby local clustering of cell surface molecules on a postsynaptic neuron dictates synapse position (*Ango et al., 2004*; *Colón-Ramos et al., 2007*; *Klassen and Shen, 2007*; *Mizumoto and Shen, 2013*). An alternative mechanism relies on axon guidance cues to restrict pre-synaptic access to one of several acceptable postsynaptic targets (the 'guidance cue' model). Guidance cues have a well-characterized role in axon and dendrite guidance (*Chisholm et al., 2016*; *Dickson, 2002*;

*For correspondence:
cdoe@uoregon.edu

**Competing interests:** The authors declare that no competing interests exist.

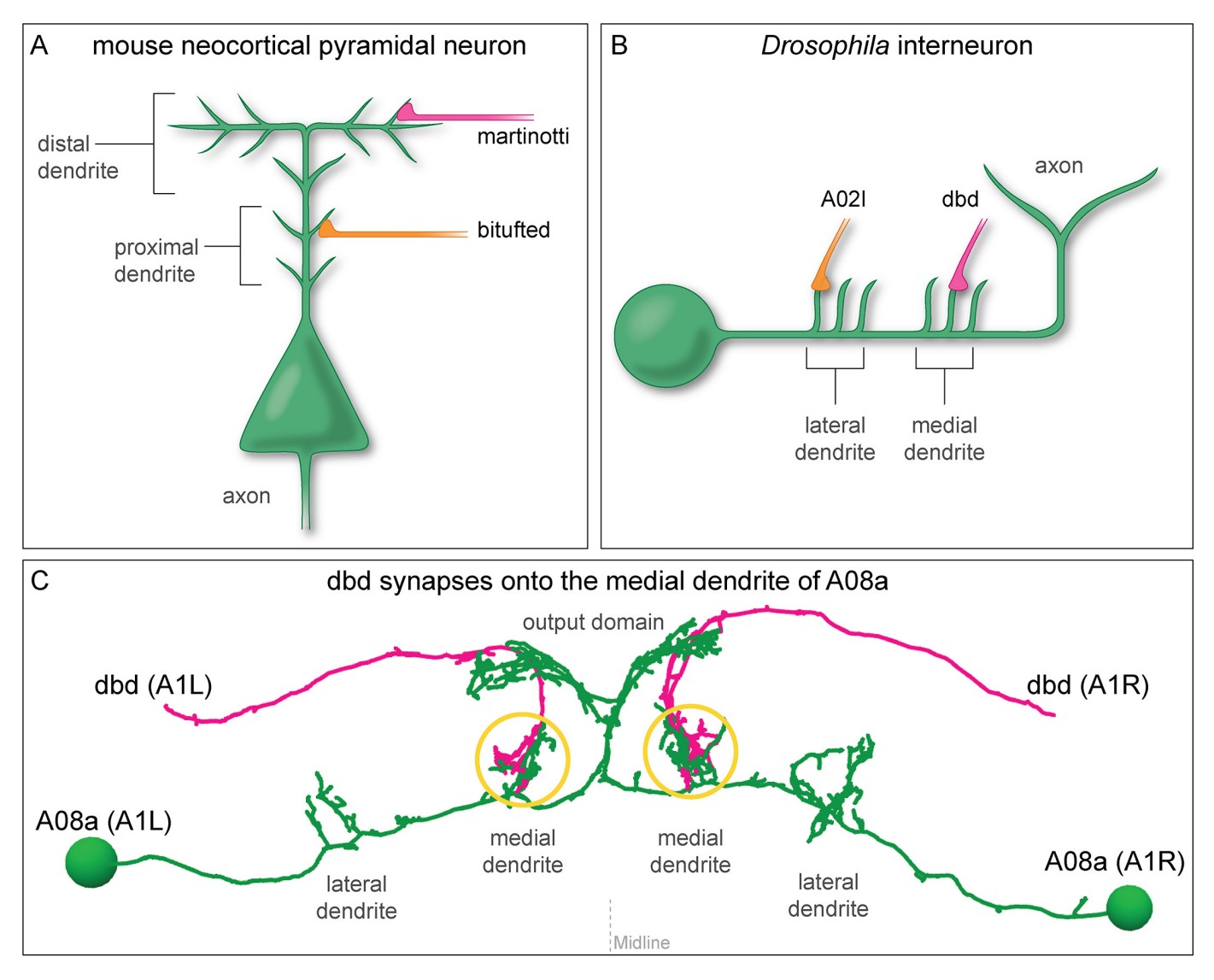

**Figure 1.** Mammalian and insect neurons display subcellular synaptic specificity. (A) Schematic of mouse neocortical pyramidal neuron (green) with a martinotti neuron (magenta) forming synapses onto the distal dendrite and the bitufted neuron (orange) forming synapses onto the proximal dendrite. (B) Schematic of fly A08a neuron (green) with a dbd neuron (magenta) forming synapses onto the medial dendrite and an A02l neuron (orange) forming synapses onto the lateral dendrite. (C) Electron microscopy reconstruction of dbd neurons (magenta) and A08a neurons (green) morphologies in one abdominal (A) segment (A1 left and A1 right) of the *Drosophila* ventral nerve cord (posterior view). dbd forms synapses specifically with the medial dendritic domain, and does not synapse with the lateral dendritic domain or the output domain.
DOI: https://doi.org/10.7554/eLife.43478.002

*Keleman and Dickson, 2001*; *Tessier-Lavigne and Goodman, 1996*; *Zlatic et al., 2009*), but their role in regulating the subcellular position of synapses has yet to be tested.

We sought to test which of these two models generate dendritic subcellular synaptic specificity using a pair of synaptically coupled neurons in the *Drosophila* larval ventral nerve cord (VNC): the dbd sensory neuron and A08a interneuron (*Itakura et al., 2015*; *Schneider-Mizell et al., 2016*). A08a has two spatially distinct dendritic arbors, one medial and one lateral, and dbd synapses specifically with the medial dendritic arbor (*Figure 1B,C*). Is this subcellular target choice due to molecular differences between the medial and lateral A08a dendritic arbors? Or are both dendritic arbors competent to accept dbd synaptic input, but axon guidance cues restrict dbd targeting to the medial arbor? Our results support the guidance cue model: we find that when the dbd axon is

lateralized in the neuropil by misexpression of the axon guidance receptor Unc-5, it forms functional synapses with the A08a lateral dendritic arbor. Taken together, our data suggest that axon guidance cues establish subcellular synaptic targeting and that there are no molecular differences in the A08a medial and lateral dendritic arbors that restrict dbd synapse formation.

## Results

### A08a interneuron has two dendritic arbors that receive distinct synaptic input

To determine which of our proposed developmental mechanisms regulates subcellular synaptic specificity, we focused on the A08a interneuron as a model system. A08a has spatially distinct medial and lateral dendrites, and receives distinct input to each of these dendrites (*Figure 2*). A08a interneurons can be visualized by light microscopy using the *R26F05(A08a)-LexA* line in larvae (24 ± 4 hr after larval hatching; ALH) in abdominal segments (A) 1–7 (*Figure 2A–A',B*). By expressing molecular markers, we determined that A08a has a distinct distal axonal (output) domain (mixed pre- and post-synapses) and a more proximal dendritic domain (predominantly post-synapses). A08a targets the dendritic marker DenMark::mCherry (*Nicolaï et al., 2010*) to the dendritic domain which includes two spatially distinct medial and lateral arbors. The A08a output domain forms a characteristic V-shaped projection at the midline, which is specifically labeled by the pre-synaptic marker Synaptotagmin::GFP (*Wang et al., 2007*) (*Figure 2C–C''*).

A08a can also be visualized by electron microscopy (EM) in first instar larvae (~5 hr ALH, *Figure 2D–D'*) (*Gerhard et al., 2017*; *Itakura et al., 2015*; *Schneider-Mizell et al., 2016*). The EM reconstruction of A08a has been completed in four hemisegments (A1 left/right, A2 left/right), and in all cases, the A08a neuron has the same arbors as seen in light microscopy: two spatially distinct dendritic arbors that contain only post-synapses, and a V-shaped output domain that contains both pre- and post-synapses (*Figure 2E*). Moreover, the same output and dendritic subcellular compartments as seen with DenMark::mCherry and Synaptotagmin::GFP can also be detected in the EM reconstructed A08a neuron using the synapse flow centrality algorithm (*Schneider-Mizell et al., 2016*), which considers path directionality between synaptic input and output locations in the A08a neuron (*Figure 2F*).

Next, we used the EM reconstruction to identify neurons with the most inputs onto A08a. We characterized the four neurons with the most synapses onto A08a dendrites (*Table 1*), and observed that dbd and A02d selectively synapse onto the A08a medial dendrite, whereas A02l and A31x selectively synapse onto the A08a lateral dendrite (*Figure 2G*; *Table 1*). Moreover, dbd-A08a partners have a synapse filling fraction similar to previously described synaptically connected neurons (*Figure 2—figure supplement 1*) (*Gerhard et al., 2017*; *Stepanyants et al., 2002*). A08a also receives synaptic input from additional neurons at its medial and lateral dendritic arbors, and these neurons also show a preference for either the medial or lateral dendritic arbor; a different set of neurons has synaptic input on the V-shaped output domain (data not shown). We conclude that the A08a neuron is an ideal model system to investigate the mechanisms generating subcellular synaptic specificity due to (a) Gal4 and LexA lines specifically expressed in A08a, (b) spatially distinct dendritic arbors with highly specific neuronal inputs onto each arbor, and (c) our ability to visualize A08a morphology by both light and electron microscopy. In addition, we have highly specific Gal4 and LexA lines for the dbd sensory neuron, which has specific synaptic input onto the A08a medial arbor (see below).

### Quantifying dbd-A08a synapse voxel position by light microscopy

The EM reconstruction allows precise quantification of synapse number and position between dbd and A08a, but EM is not a high-throughput method for experimental analysis of synaptic contacts. Thus, we developed a light microscopy method for quantifying the position of dbd-A08a putative synapse contacts. We used the genetics described above to label A08a, and additionally used the *165(dbd)-Gal4* line (*Gohl et al., 2011*) to label the dbd sensory neuron in 24 ± 4 hr ALH larvae. We conclude that dbd and A08a morphology seen in light microscopy precisely matches dbd and A08a morphology seen in the EM reconstruction (*Figure 3A–B''*, *Video 1*).

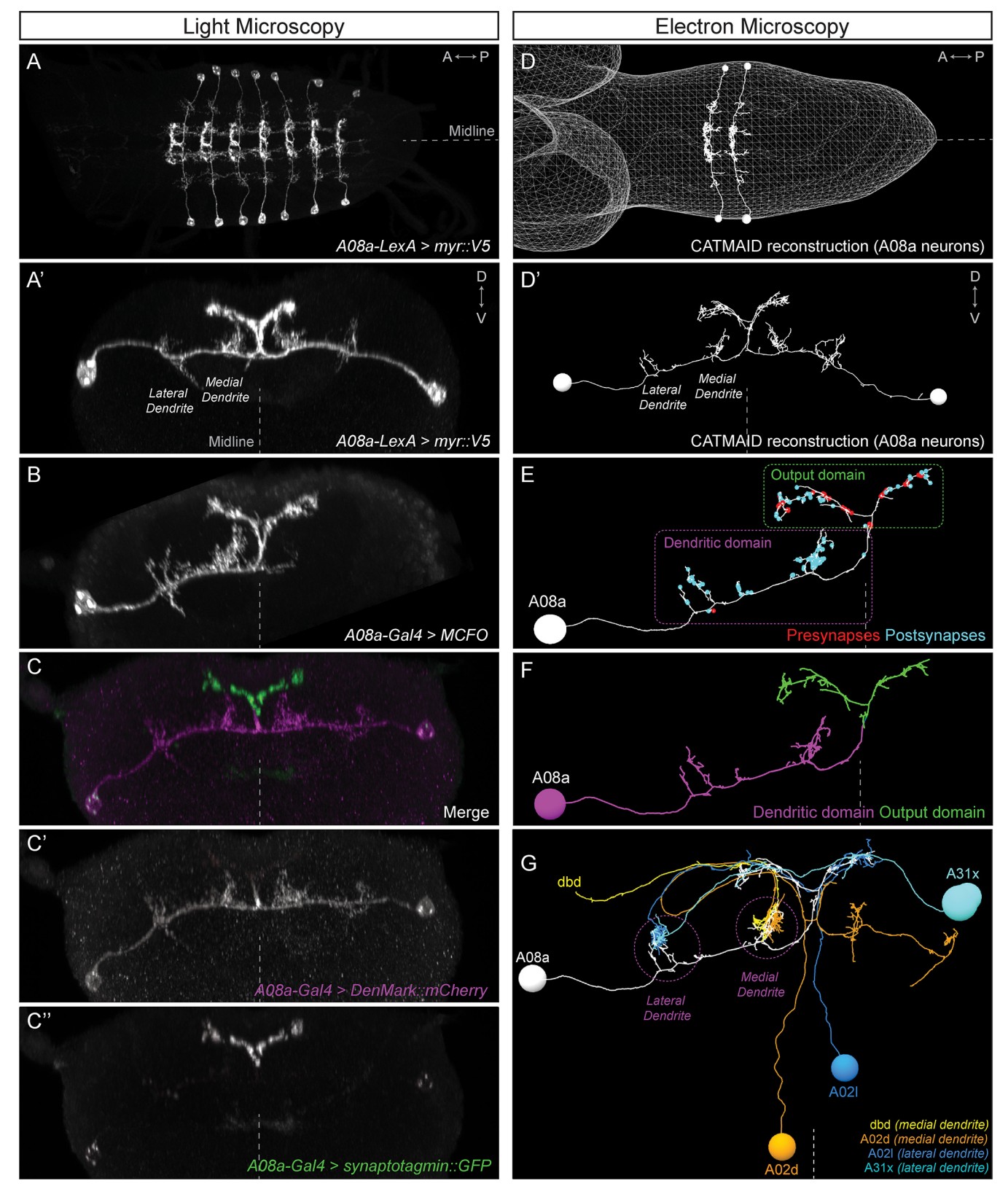

**Figure 2.** The A08a neuron receives arbor-specific synaptic inputs. (A–C'') Light microscopy (point scanning confocal) imaging of A08a neurons. (A) Dorsal view of the light micrograph (LM) 3D reconstruction of A08a neurons in the larval ventral nerve cord segments A1-7. The A08a neurons are

*Figure 2 continued on next page*

*Figure 2 continued*
visualized by *26F05(A08a)-LexA > LexAop-myr::smGdP::V5*. Midline, dashed line in all panels. (**A'**) Posterior view of the LM 3D reconstruction of paired A08a neurons in segment A1 left/right. (**B**) Posterior view of a single A08a labeled by MultiColor FlpOut (MCFO), visualized by *A08a-Gal4 > UAS-MCFO*. (**C–C''**) *A08a-Gal4* drives expression of *UAS-DenMark::mCherry* (dendrite marker) and *UAS-synaptotagmin::GFP* (presynaptic marker). Note the complementary expression in the dendritic and output domains. (**D–G**) Electron microscopy (EM) reconstruction of A08a and four synaptic partner neurons. (**D**) Dorsal view of A08a neurons in segments A1-2. (**D'**) Posterior view of A08a neurons in segment A1. (**E**) A single A08a with presynaptic and postsynaptic sites labeled in red and blue highlight a distinct dendritic domain and a mixed pre- and post-synaptic output domain, respectively. (**F**) Synapse flow centrality analyses (*Schneider-Mizell et al., 2016*) shows that A08a has distinct mixed axonal (output) and dendritic compartments. (**G**) A08a receives dendritic arbor-specific input: dbd (yellow) and A02d (orange) synapse specifically on the medial dendrite, whereas A02l (blue) and A31x (cyan) synapse specifically on the lateral dendrite.
DOI: https://doi.org/10.7554/eLife.43478.003
The following figure supplement is available for figure 2:

**Figure supplement 1.** Filling fractions between dbd and A08a neurons.
DOI: https://doi.org/10.7554/eLife.43478.004

We next quantified the position of dbd pre-synaptic contacts along the medial-lateral axis of the A08a dendrite. We used *dbd-Gal4* to express the active zone marker Bruchpilot-Short::mStrawberry (Brp-Short-mStraw, *Owald et al., 2010*) in the dbd neuron; the truncated Brp protein localizes to presynaptic sites but is not functional for inducing synapse formation, making it an excellent reporter for pre-synapses (*Fouquet et al., 2009*). In the same larvae, we used the *26F05(A08a)-LexA* line to label the A08a interneuron to express a myristoylated::V5 epitope. The dbd neuron forms synapses with many neurons in addition to A08a, so we considered only the Brp signal in close proximity (<90 nm) to the A08a membrane to define the position of dbd-A08a 'synapse voxels' (*Figure 3C–C'''*). Note that this is not designed to count individual synapse numbers, which are below the resolution limit of standard light microscopy, but rather to measure the position of putative synapses along the medio-lateral axis of the A08a dendrite. Quantifying synapse voxels across the medial-lateral axis of A08a dendrites in wild-type larvae (*Figure 3D*, n = 27 hemisegments, N = 18 animals) mirrors the position of synapses seen by EM (*Figure 3F*). In contrast, we do not observe synapse voxels between the dbd and the A08a output domain, consistent with lack of dbd synapses on the A08a output domain in the EM reconstruction (data not shown). Thus, we have established a light microscopy method for imaging and quantifying the position of dbd presynapses along the A08a dendritic membrane, which is a necessary prerequisite for investigating the mechanisms regulating dbd-A08a subcellular synaptic specificity.

**Table 1.** Summary of inputs to A08a medial and lateral dendritic arbors from the first instar larval EM reconstruction.
Neurons with the most synapses to A08a medial and lateral arbors shown. Neurons with fewer synapses also show specificity for medial or lateral dendritic arbors.

| A08a inputs (hemisegment) | Pre-synapse number | | A08a arbor targeted |
| --- | --- | --- | --- |
| | Total | With A08a | |
| dbd (A1L) | 79 | 10 | Medial only |
| dbd (A1R) | 78 | 13 | Medial only |
| A02d (A1L) | 66 | 22 | Medial only |
| A02d (A1R) | 63 | 8 | Medial only |
| A02l (A1L) | 38 | 12 | Lateral only |
| A02l (A1R) | 31 | 4 | Lateral only |
| A31x (A1L) | 19 | 3 | Lateral only |
| A31x (A1R) | 26 | 9 | Lateral only |

DOI: https://doi.org/10.7554/eLife.43478.005

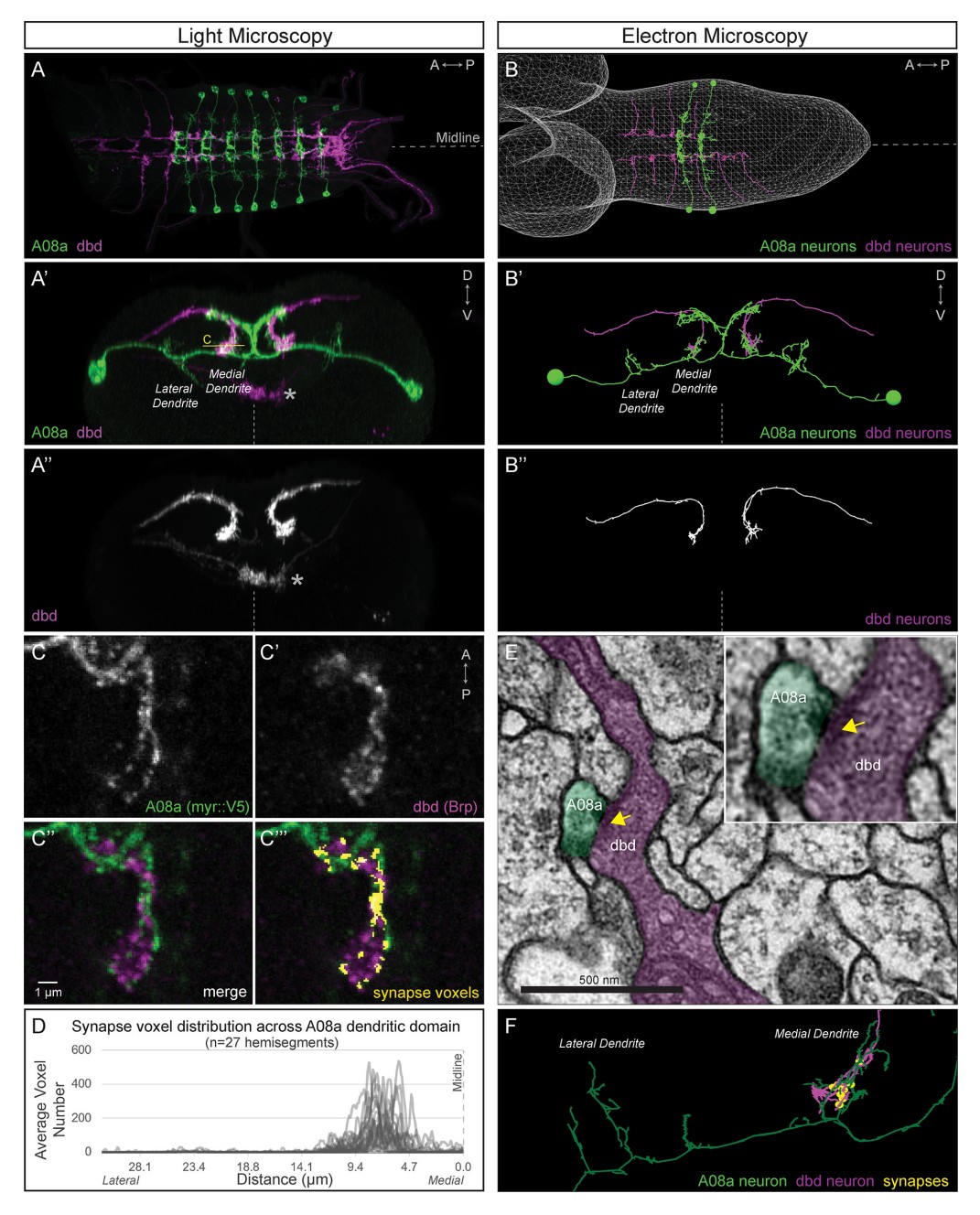

**Figure 3.** dbd and A08a neurons are synaptic partners by light and electron microscopy analyses. (**A**) Dorsal view, light microscopy 3D reconstruction showing dbd (magenta) and A08a (green) neurons. A08a is visualized with *A08a-LexA > LexAop-myr::smGdP::V5*. dbd is visualized with *dbd-Gal4 > UAS-myr::smGdP::HA*. Anterior to left; midline, dashed line in all panels. (**A'–A''**) Posterior view, light microscopy 3D reconstruction showing dbd and A08a neurons. dbd projects to the A08a medial dendritic arbor but not the A08a lateral dendritic arbor. Apparent colocalization of dbd with the A08a output domain is an artifact of the 3D projection. Asterisk, ventral off-target expression of *dbd-Gal4*. C, focal plane shown in panel C, below. (**B–B''**) EM reconstruction of dbd and A08a neurons; B, dorsal view, (A1-A2); B'-B'', posterior view, (A1). (**C–C'''**) Single optical section showing a subset of dbd presynapses (magenta, labeled with *dbd-Gal4 > UAS-brp-short-mstraw*) positioned in close proximity to the A08a membrane (green, labeled with *A08a-LexA > LexAop-myr::smGdP::V5*). Voxels containing A08a membrane within 90 nm of voxels containing Brp-mstraw are defined as 'synapse voxels' (**C'''**, yellow). (**D**) Quantification of synapse voxel position across A08a dendritic domain shows enrichment on the A08a medial dendritic arbor. (**E**) Representative chemical synapse between dbd and A08a (arrowhead) in the EM volume. (**F**) EM reconstruction showing that the dbd neuron (magenta) synapses specifically with the A08a medial but not lateral dendritic arbor (green); synapses, yellow circles.

DOI: https://doi.org/10.7554/eLife.43478.006

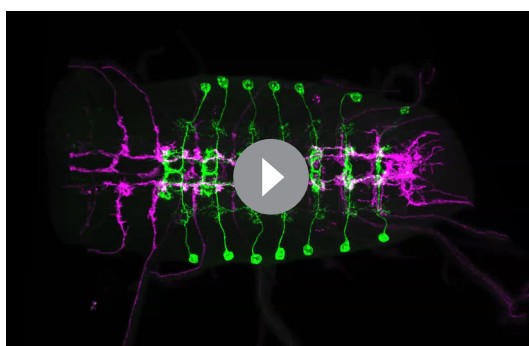

**Video 1.** dbd and A08a neurons can be visualized with light microscopy. Synaptic partners dbd (magenta) and A08a (green) can be genetically labeled (*165(dbd)-Gal4 > UAS-myr::smGdP::HA* and *26* F05(*A08a*)-*LexA > LexAop-myr::smGdP::V5* respectively).
DOI: https://doi.org/10.7554/eLife.43478.007

## Lateralized dbd has Brp + synapse voxels at the A08a lateral dendritic arbor

To determine if the lateral dendritic arbor of A08a is competent to receive input from the dbd neuron, we needed a way to re-direct dbd to a lateral location, giving it the opportunity to interact with the lateral dendrite of A08a. In *Drosophila*, neurons expressing the Netrin receptor Unc-5 or the Slit receptor Robo-2 have a repulsive response to midline-secreted Netrin and Slit ligands, respectively (*Keleman and Dickson, 2001*; *Simpson et al., 2000a*; *Simpson et al., 2000b*; *Wang et al., 2007*; *Zlatic et al., 2003*). Here, we used *dbd-Gal4* to express either Unc-5 or Robo-2 and found that both receptors could lateralize the dbd axon terminal to varying degrees, with Unc-5 being most effective and Robo-2 having a milder effect (*Figure 4—figure supplement 1*).

Wild-type dbd forms synapse voxels with the A08a medial dendritic arbor (*Figure 4A–A''*,*C*; *Figure 4—figure supplement 1B,E*). In contrast, overexpression of Unc-5 in dbd can lateralize the dbd axon terminal, positioning dbd adjacent to the A08a lateral dendritic arbor (*Figure 4B–B'*; *Figure 4—figure supplement 1D,E*). These lateralized dbd terminals formed synapse voxels with the lateral dendritic arbor of A08a (*Figure 4B''*). Similarly, overexpression of Robo-2 in dbd resulted in lateralization of the dbd axon terminal; the majority of dbd terminals formed synapse voxels in the intermediate zone between the medial and lateral dendrites (*Figure 4—figure supplement 1C,E*). The close apposition of dbd presynaptic Brp to the A08a dendritic membrane is consistent with, but does not prove, that there is functional connectivity between dbd and A08a at this arbor. Taken together, these results suggest that dbd can form Brp + putative synapses throughout the entire A08a dendritic domain, which is more consistent with the 'guidance cue' model and less consistent with the 'labeled arbor' model.

## Lateralized dbd forms functional synapses with the A08a lateral dendritic arbor

Our finding that the lateralized dbd axon terminal localizes Brp + puncta in close apposition to the lateral A08a dendritic arbor suggests that these two neurons are synaptically connected, but falls short of proving functional connectivity. To test for functional connectivity between the lateralized dbd and A08a, we took an optogenetics approach. We used the Gal4/UAS and LexA/LexAop binary expression systems (*Brand and Perrimon, 1993*; *Lai and Lee, 2006*) to express the light-gated cation channel CsChrimson (Chrimson) in dbd, and the calcium indicator GCaMP6m in A08a. For technical feasibility, all optogenetic experiments were done at the third instar larval stage (72 ± 4 hr ALH). Note that the A08a neuron at this stage retains its morphological features, including medial and lateral dendritic arbors plus a V-shaped output domain (*Figure 5—figure supplement 1*).

We first tested for functional connectivity between the wild-type dbd and A08a, which had not yet been documented. In wild-type, Chrimson-induced activation of dbd resulted in a significant increase in GCaMP6m fluorescence in A08a, but not in the absence of the Chrimson co-factor all-*trans* retinal (ATR) (*Figure 5A*, quantified in D; *Video 2*), or in the absence of the *dbd-Gal4* transgene (*Figure 5E*; quantified in F). We measured GCaMP6m levels in the output domain of A08a, which emitted a larger fluorescence signal compared to the arborizations in the dendritic domain (*Figure 5C*). This is the first experiment showing functional, excitatory connectivity between dbd and A08a. Next, we sought to determine whether the putative synapses between the lateralized dbd and the A08a lateral dendritic arbor are also functional. Using the same paradigm as in wild-type controls, we find that Chrimson activation of lateralized dbd resulted in an increase in GCaMP6m fluorescence in A08a that is statistically indistinguishable from wild-type controls (*Figure 5B*,

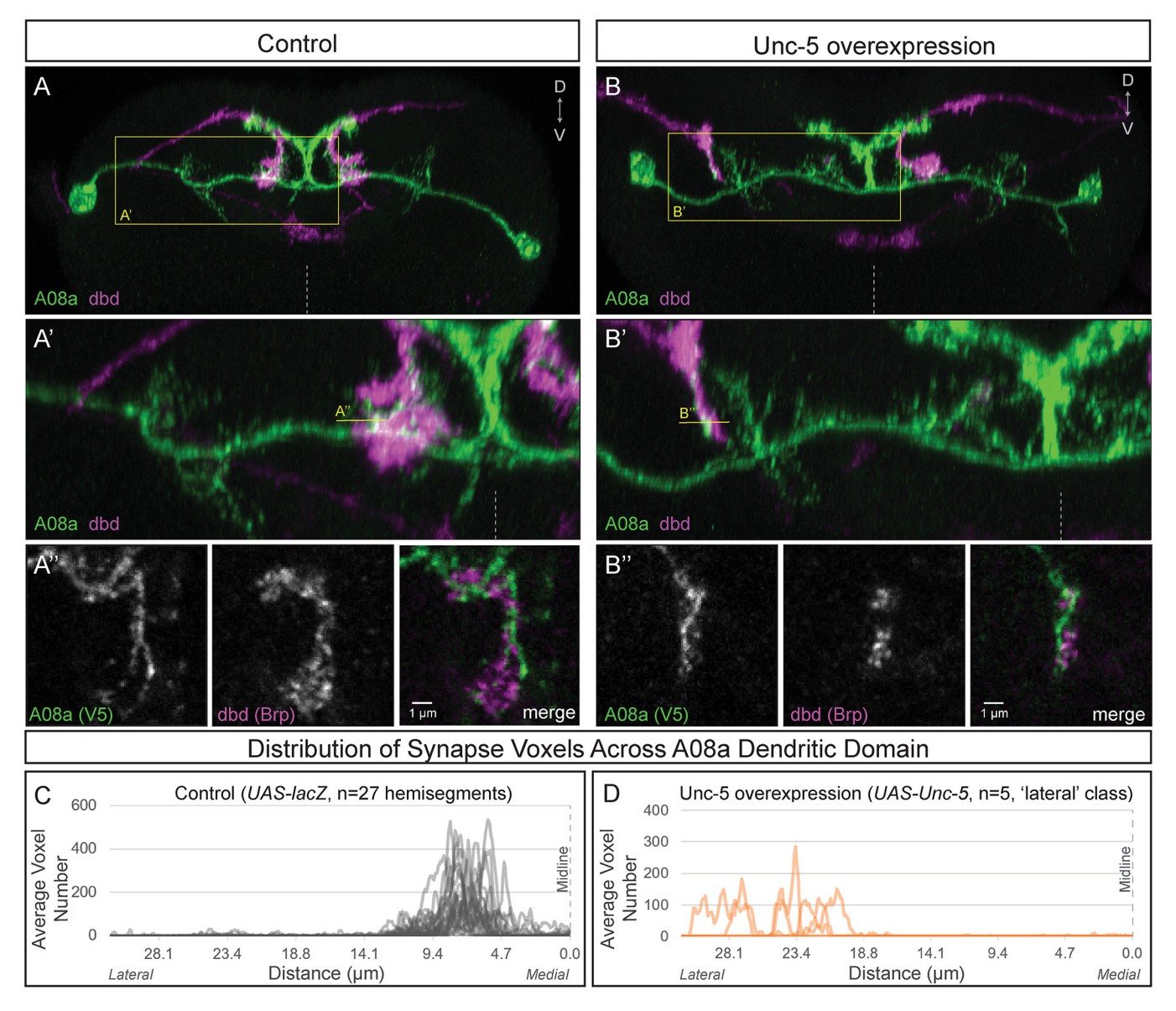

**Figure 4.** Lateralizing dbd results in Brp + putative synapses at the A08a lateral dendritic arbor. (A–A') In control animals, dbd membrane (magenta, labeled with *dbd-Gal4 >UAS-smGdP::myr::HA*) is positioned in close proximity to the A08a medial dendritic arbor membrane (green, labeled with *A08a-LexA > LexAop-myr::smGdP::V5*). (A) Posterior view of one segment; midline, dashed line in all panels; box, area enlarged in A'. (A') Posterior view of dbd and the A08a medial dendritic arbor; A'' line, optical section shown in A''. (A'') Single z-slice shows a subset of dbd presynapses (magenta, labeled with *dbd-Gal4 > UAS-brp-short::mstraw* in close proximity to the A08a medial dendritic arbor membrane. (B–B') Overexpression of Unc-5 in dbd can lateralize the axon terminal of dbd. B'' line, position of optical section shown in B'' below. See *Figure 4—figure supplement 1E* for quantification of lateralization classes. (B'') Single z-slice shows a subset of dbd presynapses (magenta, labeled with *dbd-Gal4 >UAS-brp-short::mstraw*) positioned in close proximity to A08a membrane (green, labeled with A08a-LexA > LexAop-myr::smGdP::V5). (C–D) Quantification of synapse voxel position across the dendritic domain of A08a. (C) In control animals, dbd forms synapse voxels on the medial dendritic arbor of A08a; n = 27 hemisegments from 18 animals. Data reproduced from *Figure 3D*. (D) In hemisegments with full lateralization of dbd (as shown in B'), dbd forms synapse voxels on the lateral dendritic arbor of A08a; n = 5 hemisegments from five animals. See *Figure 4—figure supplement 1E* for quantification of lateralization classes.

DOI: https://doi.org/10.7554/eLife.43478.008

The following figure supplement is available for figure 4:

**Figure supplement 1.** dbd axons can be variably lateralized by expression of axon guidance receptors Unc-5 and Robo-2.

DOI: https://doi.org/10.7554/eLife.43478.009

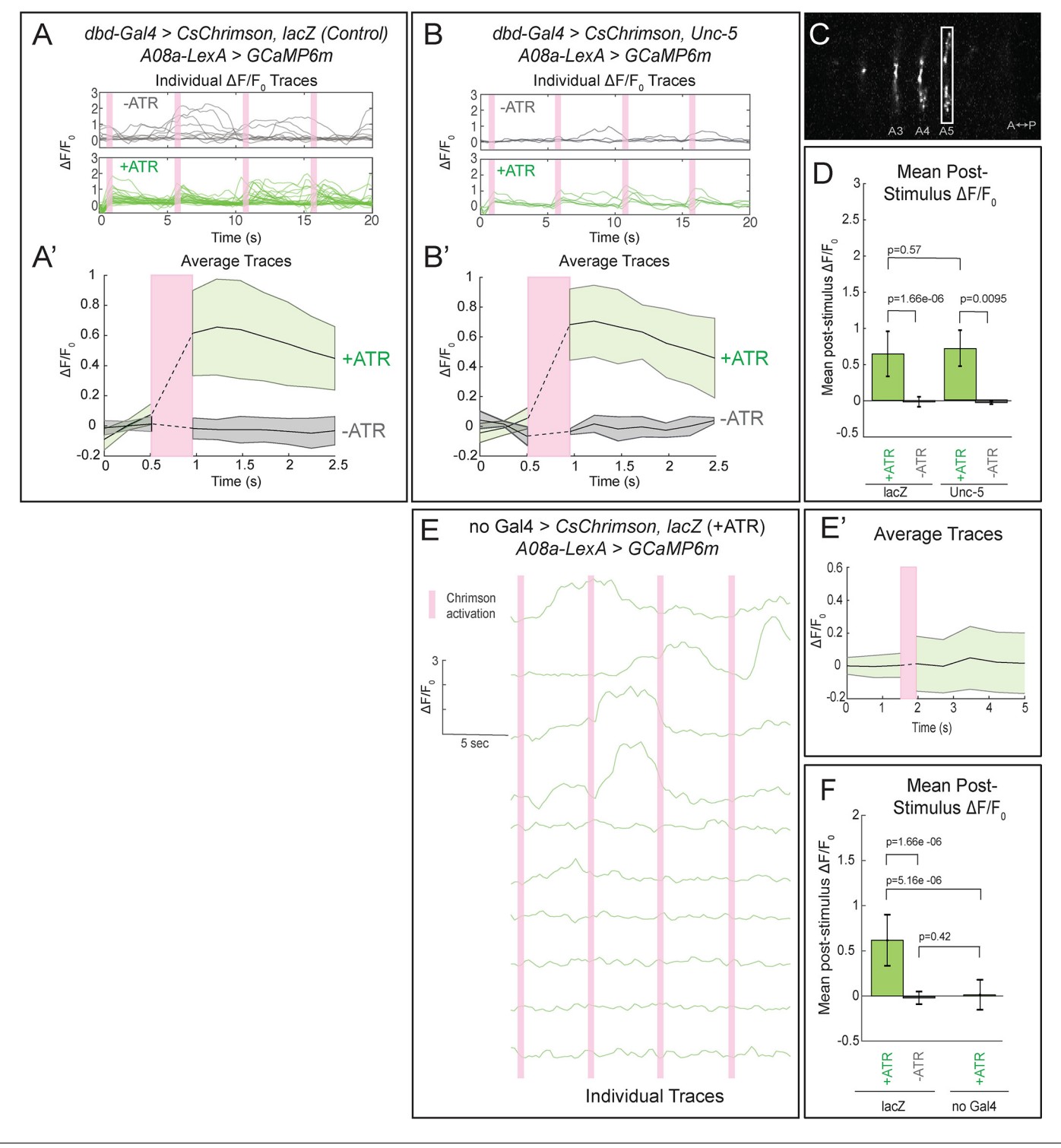

**Figure 5.** Confocal activation of Chrimson in control and lateralized dbd increases A08a GCaMP6m fluorescence. (**A–A'**) In wild-type animals, Chrimson activation of dbd neurons results in increased GCaMP6m fluorescence in the A08a output domain. For all figures,+ATR is shown in green, -ATR is shown in gray, and timing of Chrimson activation (500 ms) is represented with a pink bar. (**A**) A08a GCaMP6m $\Delta F/F_0$ traces from individual A08a pairs resulting from wild-type dbd activation. Non-evoked spontaneous activity is present in -ATR control. (**A'**) Average A08a GCaMP6m $\Delta F/F_0$ traces, before and after Chrimson activation of dbd neurons. Solid black lines represent the mean $\Delta F/F_0$. Shaded regions represent the standard deviation from the mean. +ATR, n = 28 A08a pairs, from 10 animals; -ATR, n = 11 A08a pairs, from five animals. (**B–B'**) In animals with fully lateralized dbd, Chrimson activation of dbd results in increased GCaMP6m fluorescence in A08a axon terminals. (**B**) A08a GCaMP6m $\Delta F/F_0$ traces from individual A08a pairs

*Figure 5 continued on next page*

*Figure 5 continued*

resulting from activation of lateralized dbd. (**B'**) Average A08a GCaMP6m $\Delta F/F_0$ traces, before and after Chrimson activation of dbd neurons. Solid black lines represent the mean $\Delta F/F_0$. Shaded regions represent the standard deviation from the mean. +ATR, n = 6 A08a pairs, from five animals; -ATR, n = 4 A08a pairs, from three animals. (**C**) Example ROI used for quantification drawn around A08a axon terminals in segment A5. (**D**) Quantification of the mean post-stimulus $\Delta F/F_0$ for *lacZ* control and *unc-5*. Error bars represent the standard deviation from the mean. Mean post-stimulus $\Delta F/F_0$: *lacZ* Control +ATR, 0.62 ± 0.28, n = 28 A08a pairs, from 10 animals; *lacZ* Control -ATR, −0.0172 ± 0.07, n = 11 A08a pairs, from five animals; *unc-5* +ATR, 0.68 ± 0.24, n = 6 A08a pairs, from five animals; *unc-5* -ATR, −0.035 ± 0.02, n = 4 A08a pairs, from three animals. (**E–E'**) *dbd-Gal4* is required to produce Chrimson-evoked responses in A08a. A08a expresses GCaMP6m in a genetic background containing *UAS-lacZ* and *20XUAS-CsChrimson*. (**E**) A08a GCaMP6m $\Delta F/F_0$ traces from individual A08a pairs. (**E'**) Average A08a GCaMP6m $\Delta F/F_0$ traces before and after light stimulus (pink bar). Solid black line represents the mean $\Delta F/F_0$. Shaded region represents the standard deviation from the mean. +ATR is represented in green (n = 10 A08a pairs). (**F**) Quantification of the mean post-stimulus $\Delta F/F_0$ for *lacZ* control +ATR, *lacZ* control -ATR, and no dbd-*gal4* control. Error bars represent the standard deviation from the mean. Mean post-stimulus $\Delta F/F_0$: *lacZ* Control +ATR, 0.62 ± 0.28, n = 28 A08a pairs, from 10 animals (Data reproduced from ***Figure 6D***); *lacZ* control -ATR, −0.0172 ± 0.07, n = 11 A08a pairs, from five animals (Data reproduced from ***Figure 6D***); No *dbd-gal4* Control +ATR, 0.013 ± 0.17, n = 10 A08a pairs, from five animals. Significance between two groups was determined using a Mann-Whitney test.

DOI: https://doi.org/10.7554/eLife.43478.010

The following figure supplement is available for figure 5:

**Figure supplement 1.** dbd and A08a neuronal morphology is similar at 24 hr and 72 hr after larval hatching (ALH).

DOI: https://doi.org/10.7554/eLife.43478.011

quantified in D; *Video 3*). These data are consistent with dbd activating A08a equally well using medial arbor connectivity (control) or lateral arbor connectivity (following Unc-5 expression).

We observed that the Gal4 line used to express Chrimson in dbd also has expression in a subset of ventral neurons (*Figure 3*; *Figure 5—figure supplement 1*), the stimulation of which could have hypothetically contributed to the observed A08a responses. To distinguish the influence of dbd neurons and the ventral off-targets on A08a responses, we activated each set of neurons separately via spatially restricted two-photon holographic stimulation (*Figure 6A,B*). We selected stimulation regions that were specific for each set of neurons. The stimulation regions were targeted to distinct planes and with nonoverlapping cross sections (*Figure 6C*). When we sequentially activated the dbd and off-target neurons within the same larva, we found that A08a had significantly larger GCaMP6m responses following Chrimson activation of dbd compared to the off-target neurons (*Figure 6D-F*). Similar results were observed for larvae where Unc-5 misexpression was used to lateralize the dbd axon (*Figure 6G-J*). We conclude that Chrimson activation of dbd neurons drives increased GCaMP6m fluorescence in A08a neurons in both wild-type and Unc-5 misexpression genotypes.

To determine whether the lateralized dbd provides monosynaptic input to A08a, we performed the same optogenetic experiments in the presence of tetrodotoxin (TTX), a sodium channel blocker that eliminates neuronal action potentials (*Narahashi et al., 1964*). First, we applied TTX to isolated larval CNS preparations and observed loss of the spontaneous rhythmic neuronal activity characteristic of fictive locomotion (*Pulver et al., 2015*), confirming that TTX was effective (*Figure 7A*; *Videos 4* and *5*). Next, we assayed the effect of TTX on dbd-A08a connectivity. If dbd-A08a connectivity is monosynaptic, then Chrimson activation of dbd should induce A08a GCaMP activity even in the presence of TTX; in contrast, if dbd-A08a connectivity is indirect (e.g. via feedforward excitation) then A08a GCaMP6m activity should be blocked by TTX (summarized in *Figure 7B*) (*Petreanu et al., 2009*). We found that TTX does not block dbd-induced A08a activity, in wild-type (*Figure 7C– C''*) or when the dbd axon terminal is lateralized by Unc-5 (*Figure 7D–D''*), showing that the dbd synapses on the lateral dendritic arbor of A08a are functional and monosynaptic. Interestingly,

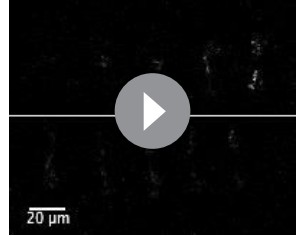

**Video 2.** Functional connectivity between dbd and A08a (*lacZ* control). Top: +ATR. A08a in WT controls exhibits stimulus-evoked changes in fluorescence. Video shows A08a axon terminals in a fictive brain preparation, anterior to the left. Bottom: -ATR. A08a does not exhibit stimulus-evoked changes in fluorescence in the absence of ATR. 'ON' indicates presentation of 561 nm light stimulus. Frames acquired at 4 frames/s, displayed at 0.5x speed.

DOI: https://doi.org/10.7554/eLife.43478.012

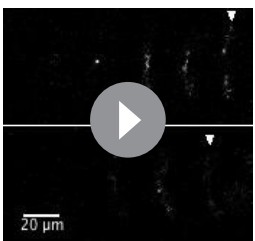

**Video 3.** Functional connectivity between lateralized dbd and A08a (*unc-5*). Top: +ATR. A08a exhibits stimulus-evoked changes in fluorescence to lateralized dbd. Video shows A08a axon terminals in a fictive brain preparation, anterior to the left. Bottom: -ATR. A08a does not exhibit stimulus-evoked changes in fluorescence in the absence of ATR. 'ON' indicates presentation of 561 nm light stimulus. White arrows indicate segments confirmed to have fully lateralized dbd's in both left and right hemisegments. Frames acquired at 4 frames/s, displayed at 0.5x speed.
DOI: https://doi.org/10.7554/eLife.43478.013

A08a GCaMP responses are significantly greater following TTX application in both wild-type and *unc-5* conditions; this may be due to the elimination of feedforward inhibition (see Discussion). We conclude that the lateralized dbd-A08a synapses are monosynaptic and functional. Our data therefore support a model in which axon guidance cues are the major determinants of dbd-A08a subcellular dendritic synaptic specificity.

## Discussion

### Achieving subcellular synaptic specificity

The ability of a presynaptic neuron to form synapses with a specific subcellular domain of its post-synaptic partner is well established in mammals (reviewed in *Yogev and Shen, 2014*) and has been described previously in *Drosophila*, although not at a mechanistic level. For example, the *Drosophila* giant fiber (GF) descending neuron targets a specific dendritic domain of the tergotrochanteral motor neuron, TTMn (*Borgen et al., 2017*). The transmembrane Sema-1a protein is required for both GF pathfinding to the motor neuropil, but also for establishing synaptic contact with the TTMn (*Godenschwege et al., 2002*; *Godenschwege and Murphey, 2009*). However, it remains unknown if Sema-1a protein is restricted to the specific dendritic domain of TTMn chosen by the GF, as predicted by the 'labeled arbor' model. Similarly, the Jaam1 and Jaam3 interneurons

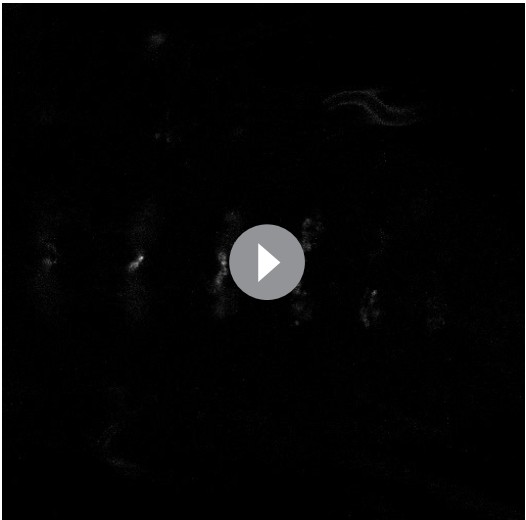

**Video 4.** Spontaneous A08a rhythmic activity (*lacZ* control, -TTX, -ATR). A08a exhibits spontaneous rhythmic activity in the absence of TTX. Video shows A08a axon terminals in a fictive brain preparation, anterior to the left. Video was acquired at two frames/ second and recorded over 5 min. Video is displayed at 2x speed.
DOI: https://doi.org/10.7554/eLife.43478.016

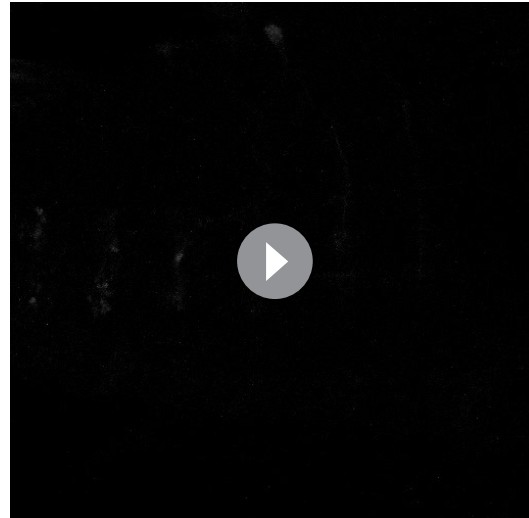

**Video 5.** TTX abolishes spontaneous A08a rhythmic activity (*lacZ* control,+TTX, -ATR). A08a spontaneous rhythmic activity is eliminated in the presence of 3 µM TTX. Video shows A08a axon terminals in a fictive brain preparation, anterior to the left. Video was acquired at two frames/second and recorded over 5 min. Video is displayed at 2x speed.
DOI: https://doi.org/10.7554/eLife.43478.017

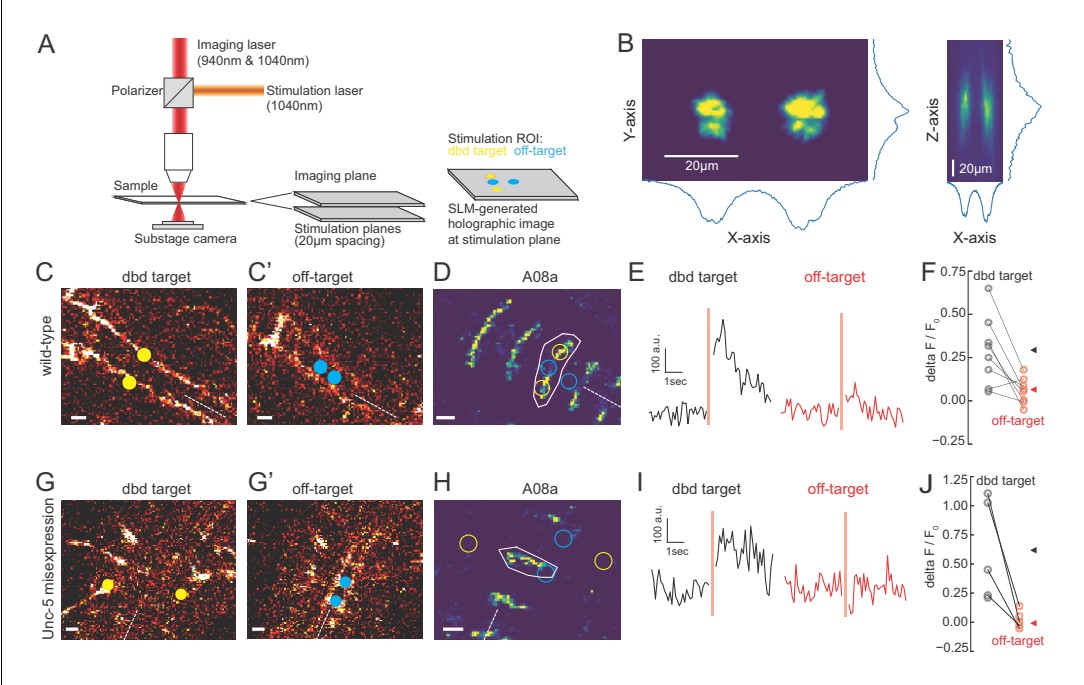

**Figure 6.** Two photon activation of dbd, but not off-target neurons, increases A08a GCaMP6m fluorescence. (A) Schematic of two photon microscope used for $Ca^{+2}$ imaging and holographic photostimulation. We used a separate imaging (940 or 1040 nm) and stimulation laser (1040 nm). Holographic photostimulation patterns were constructed with a spatial light modulator (SLM). Stimulation targeted either dbd neurons (yellow circles) or off-target neurons (blue circles), separated on average by 20 µm in the z-axis. (B) XY and XZ profile of fluorescence induced by a holographic stimulation pattern consisting of two 10 µm diameter circles separated center-to-center by 26 µm. Profiles were obtained by moving objective (and therefore stimulation pattern) systematically relative to a fixed slide with a ~1 µm fluorescent coating while imaging with a sub-stage camera. Blue lines indicate fluorescence summed across respective axes (arbitrary units). (C–F) Targeting of Chrimson stimulation and $Ca^{+2}$ imaging of A08a neurons in wild-type 72 hr ALH larvae. (C–C') Two photon image (1040 nm) of fluorescent mCherry marker at two imaging planes 20 µm apart. Stimulation ROIs used for targeting dbd (C, yellow dots) and off-target (C', cyan dots) neurons are overlaid. Dashed white line indicates midline. Scale bars, 10 µm. (D) Summed GCaMP6m fluorescence in A08a neurons (940 nm). White polygon depicts spatial region used to quantify fluorescence for traces in E. The stimulation regions shown in C are overlaid (outlines: yellow, dbd; cyan, ventral off-targets). Scale bars, 10 µm. (E) Example $Ca^{+2}$ responses from the wild-type larva shown in C,D. Black trace shows raw A08a fluorescence (arbitrary units) prior to and following 150 ms holographic stimulation of dbd targets. Red trace shows A08a fluorescence in response to ventral off-target stimulation. Stimulation timing depicted with pink rectangle. (F) Mean $Ca^{+2}$ responses ($\Delta F/F_0$) in A08a for each animal to dbd stimulation (black dots) or ventral off-target stimulation (red dots). Triangles are means for each group (dbd, 0.29 +/-. 07; off-target, 0.06 ± 0.07). N = 8 animals. Scale bars, 10 µm. (G–J) Targeting of Chrimson stimulation and $Ca^{+2}$ imaging of A08a neurons in Unc-5 misexpression larvae at 72 hr ALH. (G–G') Two photon image (1040 nm) of fluorescent mCherry marker at dbd (G) and off-target imaging planes (G'), separated by 20 µm. Stimulation ROIs overlaid (dbd, G, yellow dots; off target, G', cyan dots). (H) Summed GCaMP6m fluorescence in A08a neurons. Stimulation regions and measurement region plotted as in D. (I) Example $Ca^{+2}$ responses from Unc-5 larva shown in G, H. Plotting conventions as in E. Black trace shows raw A08a fluorescence in response to dbd stimulation; red trace is A08a fluorescence in response to off-target stimulation. (J) Mean $Ca^{+2}$ responses ($\Delta F/F_0$) in A08a for each animal to dbd stimulation (black dots) or ventral off-target stimulation (red dots). Triangles are means for each group (dbd, 0.60 ± 0.17; off-target, 0.02+/.03). N = 5 animals. Scale bars, 10 µm.

DOI: https://doi.org/10.7554/eLife.43478.014

target different domains of their post-synaptic EL neuron partners (*Heckscher et al., 2015*), but the mechanism is unknown.

Here, we provide evidence that axon guidance cues are the major determinants of subcellular dendritic synaptic specificity between dbd and A08a neurons, and that all regions of the A08a dendrite are competent to receive dbd synaptic inputs. Our findings expand upon the known mechanisms that generate subcellular synapse specificity to include guidance cues that restrict synaptic inputs to one region of a larger dendritic domain that is competent to receive synaptic input. We observed that the dbd axon is positioned close to the A08a output domain but never forms presynaptic contacts with this domain, as assayed by light and electron microscopy (data not shown). We speculate that the A08a output domain contains cell surface molecules (CSMs) that locally prevent dbd synapse formation. This is similar to work in *C. elegans* that identified secreted proteins that

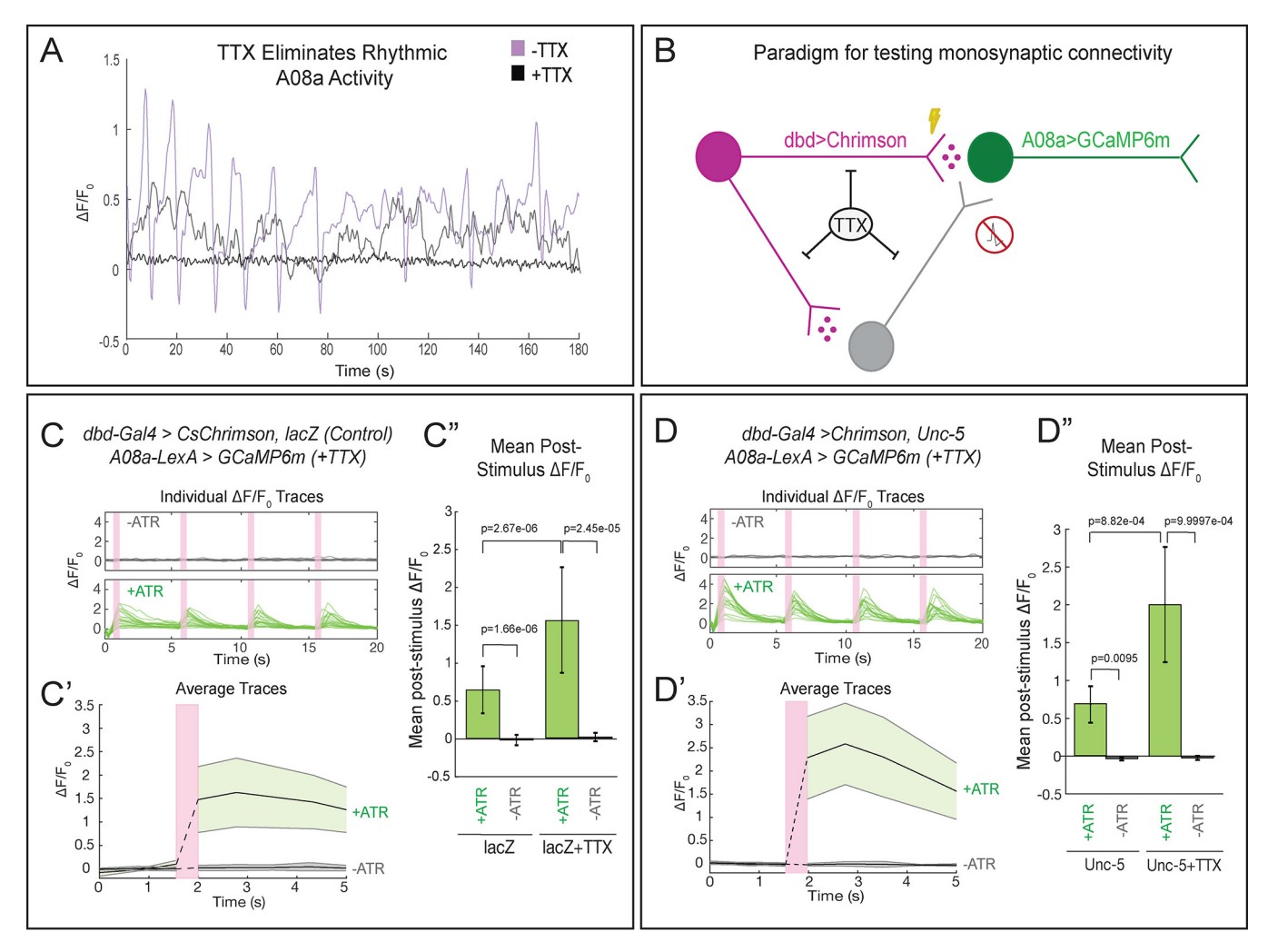

**Figure 7.** Lateralized dbd forms direct, monosynaptic connections with the A08a lateral dendrite. (**A**) TTX eliminates spontaneous rhythmic neuronal activity in A08a (in which activity is part of an inter-segmental activity wave moving in the anterior or posterior direction representing fictive motor waves; *Itakura et al., 2015*). Representative traces show the $\Delta F/F_0$ for individual pairs of A08a neurons over the course of 3 min in *lacZ* control animals. Purple trace shows A08a $\Delta F/F_0$ without TTX present. Black trace shows A08a $\Delta F/F_0$ in the presence of 3 µM TTX, in which 20/20 A08a pairs from eight animals where rhythmic activity was eliminated. In 8/20 of these A08a pairs, non-rhythmic, non-intersegmentally coordinated changes in GCaMP6m fluorescence were observed, exemplified by the gray trace (see Discussion). (**B**) Experiment to test for monosynaptic dbd-A08a connectivity. TTX eliminates action-potential-mediated activity, preventing stimulation of non-Chrimson expressing neurons. Light-activation of Chrimson induces action-potential-independent neurotransmitter release from dbd. If dbd is monosynaptically connected to A08a, increases in A08a GCaMP fluorescence will result. (**C–C"**) Wild-type dbd has excitatory, monosynaptic connection to A08a medial dendritic arbor. (**C**) A08a GCaMP6m $\Delta F/F_0$ traces from individual A08a pairs resulting from wildtype dbd activation in the presence of TTX. (**C'**) Average A08a GCaMP6m $\Delta F/F_0$ traces in the presence of 3 µM TTX, before and after Chrimson activation of dbd neurons. Solid black lines represent the mean $\Delta F/F_0$. Shaded regions represent the standard deviation from the mean. +ATR, n = 20 A08a pairs, from nine animals; -ATR, n = 9 A08a pairs, from four animals. (**C"**) Quantification of the mean post-stimulus $\Delta F/F_0$ for *lacZ* control and *lacZ* +TTX animals. Mean post-stimulus $\Delta F/F_0$: *lacZ* Control +ATR, 0.62 ± 0.28, n = 28 A08a pairs, from 10 animals (Data reproduced from *Figure 5D*); *lacZ* control -ATR, −0.0172 ± 0.07, n = 11 A08a pairs, from five animals (Data reproduced from *Figure 5D*); *lacZ* control +TTX + ATR, 1.48 ± 0.70, n = 20 A08a pairs, from nine animals; *lacZ* control +TTX ATR, 0.019 ± 0.055, n = 9 A08a pairs, from four animals. (**D–D"**) Lateralized dbd has excitatory, monosynaptic connection to A08a lateral dendritic arbor. (**D**) GCaMP6m $\Delta F/F_0$ traces from A08a pairs after activation of lateralized dbd in the presence of TTX. (**D'**) Average A08a GCaMP6m $\Delta F/F_0$ traces in the presence of 3 µM TTX, before and after Chrimson activation (pink bar) of dbd neurons. Solid black lines represent the mean $\Delta F/F_0$. Shaded regions represent the standard deviation from the mean. +ATR, n = 17 A08a pairs, from 14 animals; -ATR, n = 5 A08a pairs, from four animals. (**D"**) Quantification of the mean post-stimulus $\Delta F/F_0$ for Unc-5 and Unc-5 +TTX animals. Mean post-stimulus $\Delta F/F_0$: Unc-5 +ATR, 0.68 ± 0.24, n = 6 A08a pairs, from five animals (Data reproduced from *Figure 5D*); Unc-5 -ATR, −0.035 ± 0.02, n = 4 A08a pairs, from three animals (Data reproduced from *Figure 5D*); Unc-5 +TTX +ATR, 2.00 ± 0.76, n = 17 A08a pairs, from 14 animals; Unc-5 +TTX ATR, 0.023 ± 0.03, n = 5 A08a pairs, from four animals. Significance between two groups was determined using a Mann-Whitney test.

*Figure 7 continued on next page*

*Figure 7 continued*

DOI: https://doi.org/10.7554/eLife.43478.015

cluster CSMs to restrict synapse position on the DA9 motor neuron (*Klassen and Shen, 2007*). Similarly, NF186 expression is confined to the axon initial segment of Purkinje cells and determines the location of basket cell synapses (*Ango et al., 2004*). These observations suggest that synaptically coupled neurons may utilize both axon guidance cues and arbor-specific molecular cues to achieve subcellular synaptic specificity. We anticipate both 'labeled arbors' and 'guidance cues' play a role in determining subcellular synaptic specificity – possibly both acting in the same neuron, such as CSMs potentially regulating connectivity between coarse subcellular domains, such as the A08a axon versus dendrite, and guidance cues refining connectivity within a particular subcellular domain, such as the medial and lateral A08a dendritic domains.

## Formation of functional lateralized dbd-A08a synapses

We have shown that the lateralized dbd axon not only makes close Brp contacts with the A08a lateral dendrite, but more importantly also makes functional synapses. Interestingly, there appear to be fewer synapse voxels between the lateralized dbd and A08a than between the medial dbd and A08a, yet functional connectivity is indistinguishable. This may be due to homeostatic mechanisms that increase the efficacy of the lateral dbd-A08a synapses. The fact that the dbd-A08a optogenetic activation occurs even in the presence of TTX, together with the observation of direct dbd-A08a synapses in EM, strongly suggests that dbd and A08a have direct, monosynaptic excitatory connectivity. Interestingly, dbd induced activation of GCaMP6m in A08a is greater in the presence of TTX (in both wild-type and after dbd lateralization), suggesting that dbd may activate an inhibitory feed-forward circuit that is silenced by TTX. A good candidate for such feed-forward inhibition is the A02d neuron, which is an inhibitory neuron that receives input from dbd and has output to A08a (*Fushiki et al., 2016*; *Kohsaka et al., 2014*) (*Figure 2G*). In some cases, we detected fluctuations in A08a GCaMP6m activity following TTX application (8/20 A08a pairs; *Figure 7A*); it is unclear if these represent cases of incomplete A08a inactivation, graded $Ca^{2+}$ potentials, or $Ca^{2+}$ release from internal organelles. It is also important to consider that not all insect neurons produce sodium-dependent spikes; therefore, we cannot fully rule out the possibility that the A08a activation we observe in the presence of TTX is due to indirect stimulation from non-spiking interneurons (*Pearson and Fourtner, 1975*; *Pippow et al., 2009*).

We also note that animals fed ATR (+ATR) have a statistically significant higher baseline level of calcium activity than -ATR controls (*Videos 2* and *3* and data not shown). This is likely due to our illumination with 488 nm light between 561 nm stimulus pulses (see optogenetic Methods), because 488 nm light was shown previously to weakly activate Chrimson (*Klapoetke et al., 2014*). It therefore follows that +ATR animals would have a higher baseline level of fluorescence. Importantly, this does not change our interpretation that lateralized dbd neurons form functional synapses with the A08a lateral dendrite.

We have shown that the lateralized dbd maintains synaptic contact with A08a by remapping synaptic connectivity to the lateral arbor of A08a. However, we are unable to determine if dbd still maintains cellular synaptic specificity with its other synaptic partners. In contrast to A08a, other dbd target neurons only have a medial dendritic arbor, such as Jaam-3 (*Heckscher et al., 2015*). It would be interesting to know how these neurons respond to dbd lateralization; they may extend novel dendrite branches laterally, or may simply lose dbd synaptic inputs. The development of genetic tools to specifically label additional dbd target neurons will be required to understand if cellular synaptic specificity of dbd is maintained upon its remapping in the neuropil.

## Functional consequences of subcellular synaptic specificity

In other systems, it is well established that subcellular location of synapses has a profound impact on how a neuron propagates information within a circuit (*Bloss et al., 2016*; *Hao et al., 2009*; *Miles et al., 1996*; *Pouille et al., 2013*; *Tobin et al., 2017*). From the *Drosophila* larval EM reconstruction, we show that A08a receives distinct input into its medial and lateral dendritic arbors, which is likely to influence how A08a integrates incoming synaptic activity. dbd is a proprioceptive sensory

neuron, and A08a is rhythmically active during fictive motor waves (*Itakura et al., 2015*). Thus, the proper targeting of dbd and A02d to the medial arbor, and A02l and A31x to the lateral arbor, may be important for processing proprioceptive sensory input during locomotion. Although the *dbd-Gal4* line used in our study has ventral sensory 'off-target' expression that precludes a behavioral analysis following dbd lateralization, if this off-target expression could be removed, it is possible that the behavioral consequences of dbd lateralization could be determined using recently developed high-resolution quantitative behavior analysis tools (*Almeida-Carvalho et al., 2017*; *Kabra et al., 2013*; *Klein et al., 2017*; *Risse et al., 2017*). Furthermore, future electrophysiological studies could directly test the functional consequences of the subcellular positioning of A08a inputs on neural processing (e.g. dendritic integration, coincidence detection, and noise suppression).

# Materials and methods

## Key resources table

| Reagent type (species) or resource | Designation | Source or reference | Identifiers | Additional information |
|---|---|---|---|---|
| Species (*Drosophila melanogaster*) | 26F05-LexA | BDSC | 54702 | Expressed in A08a neurons |
| Species (*D. melanogaster*) | 26F05-Gal4 | BDSC | 49192 | Expressed in A08a neurons |
| Species (*D. melanogaster*) | 165-Gal4 | W. Grueber | N/A | Expressed in dbd neurons |
| Species (*D. melanogaster*) | UAS-LacZ | BDSC | 8529 | Control transgene |
| Species (*D. melanogaster*) | UAS-LacZ | BDSC | 8530 | Control transgene |
| Species (*D. melanogaster*) | UAS-unc-5::HA | B. Dickson | N/A | UAS drives unc-5 |
| Species (*D. melanogaster*) | UAS-robo-2::HA | BDSC | 66886 | UAS drives robo-2 |
| Species (*D. melanogaster*) | UAS-bruchpilot (short)-mstrawberry | S. Sigrist | N/A | UAS drives fluorescently labeled truncated bruchpilot |
| Species (*D. melanogaster*) | 10xUAS-IVS-myr::smGdP::HA, 13xLexAop2-IVS-myr::smGdP::V5 | BDSC | 64092 | UAS drives HA membrane tag, LexAop drives V5 membrane tag |
| Species (*D. melanogaster*) | UAS-MCFO | BDSC | 64090 | UAS drives multi-colored-flip-out |
| Species (*D. melanogaster*) | UAS-DenMark, UAS-syt.eGFP | BDSC | 33064 | UAS drives DenMark, UAS drives synaptotagmin::GFP |
| Species (*D. melanogaster*) | 13XLexAop2-IVS-p10-GCaMP6m, 20xUAS-Cs Chrimson-mCherry | V. Jayaraman | N/A | LexAop drives GCamp6m, UAS drives Chrimson |
| Antibody, monoclonal | Mouse anti-V5 | Invitrogen, Carlsbad, CA, | Cat. R96025, Lot 1949337 | (1:1000) |
| Antibody, polyclonal | Rabbit anti-mCherry | Novus Biologicals, Littleton, CO | Cat. NBP2-25157, Lot 102816 | (1:500) |
| Antibody, monoclonal | Rat anti-HA | Roche Holding, AG, Basel, Switzerland | Cat. 11867423001, Lot 27573500 | (1:100, after suggested dilution) |
| Antibody, monoclonal | Rat anti-OLLAS[DyLight-650] conjugated antibody | Novus Biologicals, Littleton, CO | Cat. NBP1-06713C, Lot F-090517c | (1:100) |
| Antibody, polyclonal | Chicken anti GFP | Aves Labs, Inc, Tigard, OR | Cat. GFP-1020, Lot. GFP697986 | (1:1000) |

*Continued on next page*

*Continued*

| Reagent type (species) or resource | Designation | Source or reference | Identifiers | Additional information |
|---|---|---|---|---|
| Antibody, polyclonal | Rabbit anti-mCherry | Novus Biologicals, Littleton, CO | Cat. NBP2-25157, Lot 102816 | (1:500) |
| Antibody, secondary | Alexa Fluor 488 AffiniPure Donkey Anti-Mouse IgG (H + L) | Jackson ImmunoResearch, West Grove, PA | Cat. 715-545-151 | (1:400) |
| Antibody, secondary | Rhodamine RedTM-X (RRX) AffiniPure Donkey Anti-Rabbit IgG (H + L) | Jackson ImmunoResearch, West Grove, PA | Cat. 711-295-152 | (1:400) |
| Antibody, secondary | Alexa Fluor 647 AffiniPure Donkey Anti-Rat IgG (H + L) | Jackson ImmunoResearch, West Grove, PA | Cat. 712-605-153 | (1:400) |
| Antibody, secondary | Alexa Fluor 488 AffiniPure Donkey Anti-Chicken IgY (IgG) (H + L) | Jackson ImmunoResearch, West Grove, PA | Cat. 703-545-155 | (1:400) |

## Fly stocks

All flies were raised at 25°C on standard cornmeal fly food.

| Genotypes | Figure |
|---|---|
| Females containing *10xUAS-IVS-myr::smGdP::HA, 13xLexAop2-IVS-myr::smGdP::V5* (BDSC# 64092); *GMR26F05-LexA* (A08a neurons) (BDSC# 54702), *UAS-bruchpilot (short)-mstraw;* *165* Gal4 (dbd neurons) were crossed to males containing *UAS-lacZ.Exel* (control) (BDSC# 8529) | *Figure 2A-A';* *Figure 3A-A'';* *Figure 3C-D;* *Figure 4A-A'',* *Figure 4C,* *Figure 4— figure supplement 1B, E;* |
| Females containing *10xUAS-IVS-myr::smGdP::HA, 13xLexAop2-IVS-myr::smGdP::V5* (BDSC# 64092); *GMR26F05-LexA* (A08a neurons) (BDSC# 54702), *UAS-bruchpilot(short)-mstraw; 165* Gal4 (dbd neurons) were crossed to males containing *UAS-robo-2::HA* (BDSC# 66886) | *Figure 4— figure supplement 1C, E* |
| Females containing *10xUAS-IVS-myr::smGdP::HA, 13xLexAop2-IVS-myr::smGdP::V5* (BDSC# 64092); *GMR26F05-LexA* (A08a neurons) (BDSC# 54702), *UAS-bruchpilot(short)-mstraw; 165* Gal4 (dbd neurons) were crossed to males containing *UAS-unc-5::HA* | *Figure 4B-B'';* *Figure 4D;* *Figure 4— figure supplement 1D, E* |
| Females containing *GMR57C10-FLPL;; 10xUAS(FRT.stop) myr::smGdP-OLLAS, 10xUAS(FRT.stop)myr::smGdP::HA, 10xUAS(FRT.stop)myr::smGdP::V5-THS-10xUAS (FRT.stop)myr::smGdP-FLAG* (MCFO) (BDSC# 64090) were crossed to males containing *GMR26F05-Gal4* (BDSC# 49192) | *Figure 2B* |
| Females containing *GMR26F05-Gal4* (BDSC# 49192) were crossed to males containing *UAS-DenMark, UAS-syt.eGFP; In(3L)D, mirr/TM6C, Sb* (BDSC# 33064) | *Figure 2C-C''* |

*Continued on next page*

*Continued*

| Genotypes | Figure |
|---|---|
| Females containing *GMR26F05-LexA* (BDSC# 54702); *165 Gal4* were crossed to males containing *UAS-lacZ.Exel*; *13XLexAop2-IVS-p10-GCaMP6m*, *20xUAS-CsChrimson-mCherry* (control) | *Figure 5A-A'*; *Figure 5D'*; *Figure 5F*; *Figure 6C-F*; *Figure 7A*; *Figure 7C-C'* |
| Females containing *GMR26F05-LexA* (BDSC# 54702); *165-Gal4, UAS-unc-5::HA* were crossed to males containing *13XLexAop2-IVS-p10-GCaMP6m*, *20xUAS-CsChrimson-mCherry* | *Figure 5B-B'*; *Figure 5D*; *Figure 6G-J*; *Figure 7D-D'* |
| Females containing *UAS-lacZ.Exel*; *13XLexAop2-IVS-p10-GCaMP6m*, *20xUAS-CsChrimson-mCherry* were crossed to males containing *GMR26F05-LexA* (BDSC# 54702) (No Gal4 control) | *Figure 5E-F* |

## Immunohistochemistry and sample preparation

### Larval preparation

Collection of timed larvae: embryos and larvae were raised at 25˚C. Embryos were collected on 3.0% agar apple juice caps with yeast paste for 4 hr and then aged for 21 hr. Embryos were transferred to a fresh 3.0% agar apple juice cap and then aged for 4 hr. Hatched larvae were transferred to standard cornmeal fly food vials and aged until dissection.

### Immunohistochemistry

Larval brains were dissected in PBS, mounted on 12 mm #1.5 thickness poly-L-lysine coated coverslips (Neuvitro Corporation, Vancouver, WA, Cat# H-12–1.5-PLL) and fixed for 23 min in fresh 4% paraformaldehyde (PFA) (Electron Microscopy Sciences, Hatfield, PA, Cat. 15710) in PBST. Brains were washed in PBST and then blocked with 2.5% normal donkey serum and 2.5% normal goat serum (Jackson ImmunoResearch Laboratories, Inc, West Grove, PA) in PBST overnight. Brains were incubated in primary antibody for two days at 4˚C. The primary was removed and the brains were washed with PBST, then incubated in secondary antibodies overnight at 4˚C. The secondary antibody was removed following overnight incubation and the brains were washed in PBST. Brains were dehydrated with an ethanol series (30%, 50%, 75%, 100%, 100%, 100% ethanol; all v/v, 10 min each) (Decon Labs, Inc, King of Prussia, PA, Cat. 2716GEA) then incubated in xylene (Fisher Chemical, Eugene, OR, Cat. X5-1) for 2 × 10 min. Samples were mounted onto slides containing DPX mountant (Millipore Sigma, Burlington, MA, Cat. 06552) and cured for 3 days then stored at 4˚C until imaged.

The following primary and secondary antibodies were used:

| Primary antibody (concentration) | Source | Figure |
|---|---|---|
| Mouse anti-V5 tag monoclonal antibody (1:1000) Rabbit anti-mCherry polyclonal antibody (1:500) Rat anti-HA tag monoclonal antibody (1:100, after suggested dilution) | Invitrogen, Carlsbad, CA, Cat. R96025, Lot 1949337 Novus Biologicals, Littleton, CO, Cat. NBP2-25157, Lot 102816 Roche Holding, AG, Basel, Switzerland, Cat. 11867423001, Lot 27573500 | *Figure 2A-A'*; *Figure 3A-A''*; *Figure 3C-C'''*; *Figure 4A-B''*; *Figure 4— figure supplement 1B-D* |

*Continued on next page*

*Continued*

| Primary antibody (concentration) | Source | Figure |
|---|---|---|
| Rat anti-OLLAS[DyLight-650] conjugated antibody (1:100) | Novus Biologicals, Littleton, CO, Cat. NBP1-06713C, Lot F-090517c | *Figure 2B* |
| Chicken anti GFP polyclonal antibody (1:1000) (labels Syt:GFP) Rabbit anti-mCherry polyclonal antibody (1:500) (labels DenMark) | Aves Labs, Inc, Tigard, OR, Cat. GFP-1020, Lot. GFP697986 Novus Biologicals, Littleton, CO, Cat. NBP2-25157, Lot 102816 | *Figure 2C-C''* |

| Secondary antibody (concentration) | Source | Figure |
|---|---|---|
| Alexa Fluor 488 AffiniPure Donkey Anti-Mouse IgG (H + L) (1:400) Rhodamine RedTM-X ( RRX) AffiniPure Donkey Anti-Rabbit IgG (H + L) (1:400) Alexa Fluor 647 AffiniPure Donkey Anti-Rat IgG (H + L) | Jackson ImmunoResearch, West Grove, PA, Cat. 715-545-151 Jackson ImmunoResearch, West Grove, PA, Cat. 711-295-152 Jackson ImmunoResearch, West Grove, PA, Cat. 712-605-153 | *Figure 2A-A'*; *Figure 3A-A''*; *Figure 3C-C'''*; *Figure 4A-B''*; *Figure 4— figure supplement 1B-D* |
| Alexa Fluor 488 AffiniPure Donkey Anti-Chicken IgY (IgG) (H + L) (1:400) Rhodamine RedTM-X (RRX) AffiniPure Donkey Anti-Rabbit IgG (H + L) (1:400) | Jackson ImmunoResearch, West Grove, PA, Cat. 703-545-155 Jackson ImmunoResearch, West Grove, PA, Cat. 711-295-152 | *Figure 2C-C''* |

## Light microscopy

Fixed larval preparations were imaged with a Zeiss LSM 800 laser scanning confocal (Carl Zeiss AG, Oberkochen, Germany) equipped with an Axio Imager.Z2 microscope. A 63x/1.40 NA Oil Plan-Apochromat DIC m27 objective lens and GaAsP photomultiplier tubes were used. Software program used was Zen 2.3 (blue edition) (Carl Zeiss AG, Oberkochen, Germany). For each experiment, all samples were acquired using the same acquisition parameters (see below).

| Voxel size | Excitation wavelength (laser power) | Detection wavelength | Pinhole size (AU) | Figure |
|---|---|---|---|---|
| 0.090 × 0.090×0.280 µm³ | 488 nm (0.13%) 561 nm (0.07%) 640 nm (0.14%) | 410–541 nm 541–627 nm 656–700 nm | 35 µm for all channels (488 nm: 0.82AU, 561 nm: 0.71AU, 647 nm: 0.63AU) | *Figure 2A-A'*; *Figure 3A-A''*; *Figure 3C-C'''*; *Figure 4A-B''*; *Figure 4— figure supplement 1B-D* |
| 0.067 × 0.067×0.280 µm³ | 640 nm (0.65%) | 656–700 nm | 40 µm (0.72AU) | *Figure 2B* |
| 0.067 × 0.067×0.280 µm³ | 488 nm (0.13%) 561 nm (0.25%) | 410–540 nm 540–772 nm | 43 µm (0.99AU) 38 µm (0.77AU) | *Figure 2C-C''* |

## Image processing and analyses
### Quantification of dbd-A08a synapse voxel distribution

The 'synapse voxel' image analyses pipeline identifies Brp voxels that are either one voxel away or already overlapping with membrane containing voxels. Since each voxel size is 90 nm, then the 'synapse voxels' represent the voxels that have Brp less than 90 nm away from membrane voxels.

Image processing and analysis was performed using FIJI (ImageJ 1.50d, https://imagej.net/Fiji). Stepwise, images were rotated (Image >Transform > Rotate(bicubic)) to align A08a dendrites along the X-axis, then a region of interest was selected in 3D to include A08a dendrites in one hemi-segment (Rectangular selection >Image > Crop). The Brp and A08a dendrite channels were isolated (Image >Color > Split channels). To quantify the amount of voxels containing A08a dendrite signal within 90 nm of voxels containing Brp signal, a mask was manually applied to each channel (Image >Adjust > Threshold). The threshold was assigned to include Brp positive voxels and minimize contribution from background. Because of the inherent variability in pixel intensity between different samples (most likely due to the variability of the Gal4 and LexA systems), we could not assign the same threshold to different samples. We found that manually assigning thresholds was a more accurate method of identifying Brp or membrane containing voxels compared to automatic thresholding methods available in FIJI. Importantly, the Brp and membrane thresholds were assigned separately and prior to quantifying the number of overlapping voxels.The Brp mask channel was dilated one iteration (Process >Binary > Dilate). We assigned the 90 nm distance threshold to account for the size of the synaptic cleft (~20 nm, measured in EM) and the chromatic aberration between 488 nm and 555 nm wavelengths used to visualize A08a membrane and dbd presynapses (~70 nm, measured in our light microscope). Then image arithmetic was used to identify the voxels that contain intensity in both the masked A08a dendrite and dilated Brp channels (Process >Image Calculator >Operation 'AND'). Images were reduced in the z-dimension (Image >Stacks > Z-project>Sum Slices) and a plot profile was obtained to measure the average voxel intensity across the medial-lateral axis of A08a dendrites (Rectangular selection >Analyze > Plot profile). Distance from the midline was calculated by setting a starting point at the midline and then calculating distance along the medio-lateral axis perpendicular to the midline.

Filling fractions were defined as previously described (*Gerhard et al., 2017*).

## Figure preparation

Images in figures were prepared as either 3D projections in Imaris 9.2.0 (Bitplane AG, Zurich, Switzerland) or maximum intensity projections in FIJI (ImageJ 1.50d, https://imagej.net/Fiji). Scale bars are given for reference on maximum intensity projections and single z-slice micrographs, but do not necessarily represent actual distances, as the tissue samples undergo changes in size during the tissue clearing protocol. Pixel brightness was adjusted in some images for better visualization; all such adjustments were made uniformly over the entire image.

Scale bars were included in all single focal planes and standard maximum intensity projections. In some cases, figures were '3D projected' images exported from the Imaris software, where the scale bars are assigned to match the scale at the 'center' of the 3D projection. In these cases we did not add a scale bar because it would not be accurate for all parts of the image.

## Data collection

A power analysis was not performed to determine the appropriate sample size. Many samples were dissected to account for low penetrance of dbd lateralization and to account for damaged samples that were not suitable for image analyses. All sample numbers represent biological replicates. However, we did perform the same experiment on multiple days. We did not exclude any outliers from the data sets. The criteria for excluding samples were as follows. For the fixed tissue preparation, samples with poor dissection quality or poor mounting on slides were excluded as they were unsuitable for the image analyses pipeline. Samples were also excluded if random 'off-target' neuron expression interfered with image analysis. For optogenetic experiments, samples were excluded if sample movement in the z-axis precluded accurate quantification of changes in fluorescence. For lateralized dbd optogenetics, brain segments were excluded from analysis if A08a received input from dbd on the medial dendrite. Samples were allocated into groups by genotype; every genotype was treated as an independent group.

## Functional connectivity assays

Newly hatched larvae were aged for 48 ± 4 hr ALH on standard cornmeal fly food at 25°C. At this time, larvae were transferred to apple caps containing wet yeast supplemented with 0.5 mM all-*trans* retinal (Sigma-Aldrich, R2500-100MG) and aged at 25°C in the dark. Following another 24 hr (72 ± 4

hr ALH) animals were dissected in HL3.1 saline solution. All dissections were performed in low lighting to prevent premature Chrimson activation. Freshly dissected brains were mounted in HL3.1 saline on 12 mm round Poly-L-Lysine-coated coverslips.

## Confocal experiments (*Figures 5* and *7*)

GCaMP6m signal in postsynaptic A08a axon terminals was imaged using 0.01% power of the 488 nm laser with a 40x objective on a Zeiss LSM800 confocal microscope (NA: 1.4; pinhole size: 32 μm (1AU); detection wavelength: 450–550 nm, voxel size: $0.782 \times 0.782 \times 1$ μm$^3$). Chrimson in presynaptic neurons was activated with three pulses of 561 nm laser at 100% power delivered via the same 40x objective using the bleaching function in the ZEN Zeiss software. The total length of the 561 nm pulses was about 450msec. After individual recording sessions of *unc-5* expressing samples, Z-stacks of the brain were taken to verify the segments in which A08a exclusively received dbd input onto the lateral dendrite and were therefore permissible for analysis; the few larvae where Chrimson + off target neurons were close to A08a neurons were excluded, although due to low signal we can't exclude the possibility of rare or fine contacts. A08a neurons from abdominal segments 3–5 were used for our analyses, as no statistically significant difference in post-stimulus $\Delta F/F_0$ was detected among these neurons.

To quantify $\Delta F/F_0$ traces we used a custom MATLAB script (The MathWorks, Natick, MA). The script first performs rigid registration to correct for movement artifacts during recording, and then allows for ROI selection. ROIs were drawn around A08a axon terminals in individual segments, and ROI size was constant across all experiments (*Figure 5C*). $F_0$ was set as the average fluorescence of the three frames acquired before each 561 nm light stimulus. For a single animal, we first average $\Delta F/F_0$ traces for six consecutive 561 nm stimuli separated by 20 488 nm acquisition frames (four frames/sec). These 20 frames are enough time to allow GCaMP6m fluorescence to return to baseline. Traces were then averaged across animals to determine the mean $\Delta F/F_0$ for each experimental group. Mean post-stimulus $\Delta F/F_0$ was calculated by first subtracting the mean $F_0$ from the mean F in the first frame post-stimulus, then dividing the resulting $\Delta F$ by the mean $F_0$. The mean was then calculated for each experimental group.

For demonstrating monosynaptic connectivity between dbd and A08a, brains were dissected and mounted in 3 μM TTX (Abcam, Cambridge, MA, ab120055) diluted in HL3.1. Brains were incubated for 5 min in the TTX solution prior to the recording session. To first determine the effectiveness of TTX, spontaneous A08a GCaMP6m activity was recorded over 5 min with and without TTX (in *lacZ* control animals). Spontaneous GCaMP6m activity was recorded on an LSM800 with a 40X objective (NA: 1.4; excitation wavelength: 488 nm; detection wavelength: 492–555 nm; pinhole size: 32 μm (1AU)). Once it was established that TTX eliminates spontaneous rhythmic A08a activity, we dissected fresh brains in TTX and performed the same Chrimson activation paradigm (using the same bleaching protocol and image acquisition settings) as described above to test monosynaptic connectivity.

## Two photon experiments (*Figure 6*)

Images were generated using a galvanometric and resonant scan mirror-based two-photon microscope (VIVO Multiphoton Movable Objective RS +Microscope and Vector resonant galvo scanner, 3i , Denver, CO). A Zeiss W Plan-Apochromat 20x/1.0 NA water dipping objective (apochromatically corrected 480 nm-1300nm) with a working distance of 2.3 mm was used for delivery of excitation and stimulation laser excitation. The imaging system utilizes the Chameleon Discovery dual wavelength laser system (Coherent, Santa Clara, CA) as the pump laser. The pump laser supplies 100 fs pulses at an 80 MHz repetition with an output power of 1.3 W at 940 nm and 3.9 W at 1040 nm. Imaging frames were obtained at a 39.6 Hz, and five frames were averaged per saved image. The scan range was 578 μm x 571 um, corresponding to a pixel size of 1.47 μm x 1.42 um. GCaMP6m and mCherry were excited using 940 nm (27 mW) and 1040 nm (200–244 mW) radiation, respectively, while the fluorescence was collected with two fast-gated GaAsP PMTs having filter sets that selectively collect fluorescence between 490 and 560 nm for the green channel and 570 and640 nm in the red channel.

Sample stimulation was based around a 5 W, 192 fs, 10 MHz laser system for excitation of Chrimson at 1040 nm (FemtoTrain 1040–5, Spectra-Physics, Santa Clara, CA). Excitation was delivered

through the objective with a phase-only spatial light modulator (SLM) (Phasor, computer—generated holography system, 3i , Denver, CO) for precise patterned and 3D photomanipulation. Between 21 mW and 66 mW were used in 150 ms stimulation pulses for Chrimson activation. Stimulation ROIs were two 10 um diameter circles localized over regions of interest guided by two-photon imaging of the mCherry marker. Holographic stimulation allowed for Chrimson activation at arbitrary depths within the sample while continuously monitoring A08a fluorescence in the imaging plane.

For quantification of $\Delta F/F_0$ responses to two-photon activation (*Figure 6*), we computed F0 as the mean fluorescence over the 20 frames (2.53 s) prior to the 150 ms stimulus. $\Delta F$ was computed as the difference between $F_0$ and the mean fluorescence over the 5 frames following the stimulus (0.63 s).

## Statistical analyses

Statistical analyses for optogenetic experiments were performed with MATLAB and R. For analyzing the statistical significance of mean post-stimulus $\Delta F/F_0$, an H-test was used to determine whether the data for each experimental group were normally distributed. Because these data were non-normally distributed, a Mann-Whitney test was performed to determine whether there were statistically significant differences in mean $\Delta F/F_0$ among experimental groups. To analyze potential differences in $F_0$ among + and - ATR groups we used a Pairwise Wilcox Test to calculate comparisons between each experimental group. This was followed by a Benjamini and Hochberg correction for multiple testing. All code for analysis of optogenetic data in *Figures 5–7* is deposited at the following GitHub repository https://github.com/timothylwarren/elife_larvae_2019 (*Warren, 2019*; copy archived at https://github.com/elifesciences-publications/elife_larvae_2019).

## Acknowledgements

We thank Larry Scatena for expert technical assistance with the two photon experiments and for panels 6A and B, Wes Grueber for providing *165*-Gal4, Stephan Sigrist for providing *UAS-brp-short:: mstraw*, Vivek Jayaraman for providing optogenetics fly stocks, and Barry Dickson for providing *UAS-unc-5* stocks. We thank Brandon Mark for MATLAB scripts and support with the CATMAID database. We thank Keiko Hirono for assistance with embryo and larval collections. We thank Casey Doe and Cooper Doe for assistance with brain dissections. We thank Aref Zarin and Avinash Khandelwal for annotating neurons in the CATMAID database. We thank lab members, Tory Herman, Shawn Lockery, Judith Eisen, Joseph Brückner, and Adam Miller for comments on the manuscript. We thank Jan Trout for model neuron illustrations in *Figure 1*. Stocks obtained from the Bloomington Drosophila Stock Center (NIH P40OD018537) were used in this study. Funding was provided by HHMI (CQD, ECS, ELH, TLW), NIH HD27056 (CQD), and the University of Oregon Developmental Biology Training Program (ELH). Emily C Sales is a Howard Hughes Medical Institute Gilliam Fellow.

## Additional information

### Funding

| Funder | Grant reference number | Author |
|---|---|---|
| Howard Hughes Medical Institute | Gilliam fellowship | Emily C Sales |
| University of Oregon | Developmental Biology Training Program | Emily L Heckman |
| Howard Hughes Medical Institute | | Chris Q Doe Emily L Heckman Timothy L Warren |
| National Institutes of Health | HD27056 | Chris Q Doe |

The funders had no role in study design, data collection and interpretation, or the decision to submit the work for publication.

## Author contributions
Emily C Sales, Conceptualization, Investigation, Methodology, Writing—original draft, Writing—review and editing, Major contributor to Figures 2-4 and associated supplemental figures; Emily L Heckman, Conceptualization, Investigation, Methodology, Writing—original draft, Writing—review and editing, Major contributor to Figures 5-7 and associated supplemental figures; Timothy L Warren, Resources, Investigation, Methodology, Writing—review and editing, Performed data analysis in Figure 6 and assembled the figure; Chris Q Doe, Conceptualization, Data curation, Supervision, Funding acquisition, Writing—original draft, Project administration, Writing—review and editing

## Author ORCIDs
Emily L Heckman (iD) http://orcid.org/0000-0002-0012-3364
Timothy L Warren (iD) http://orcid.org/0000-0002-4429-4106
Chris Q Doe (iD) http://orcid.org/0000-0001-5980-8029

## Decision letter and Author response
Decision letter https://doi.org/10.7554/eLife.43478.020
Author response https://doi.org/10.7554/eLife.43478.021

## Additional files

### Supplementary files
• Transparent reporting form
DOI: https://doi.org/10.7554/eLife.43478.022

### Data availability
Code is deposited to the following GitHub repository (https://github.com/timothylwarren/elife_larvae_2019; copy archived at https://github.com/elifesciences-publications/elife_larvae_2019).

The following datasets were generated:

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
