## [Decision Letter]

Thank you for submitting your article "Regulation of subcellular dendritic synapse specificity by axon guidance cues" for consideration by *eLife*. Your article has been reviewed by two peer reviewers, and the evaluation has been overseen by Graeme Davis as Reviewing Editor and K VijayRaghavan as the Senior Editor. The reviewers have opted to remain anonymous.

The reviewers have discussed the reviews with one another and the Reviewing Editor has drafted this decision to help you prepare a revised submission.

Summary:

In their manuscript, Sales and colleagues address the question of subcellular synaptic specificity using the *Drosophila* larval nervous system. First, they establish A08a as a model system. They clearly demonstrate, using light microscopy and reconstruction of EM volumes, that A08a has two dendritic domains (lateral and medial) that receive distinct inputs; importantly, the dbd sensory neurons rather selectively synapse on the medial domain. Using this system, they aim to test whether axon guidance cues in the sensory neuron or local cues (the molecular nature of such cues, whether they're derived from the postsynaptic neuron or elsewhere, etc. are not clearly defined in their model) determine the pattern of dbd-A08a synaptogenesis. Based on the observation that mistargeting of dbd axons to lateral domains in the neuropil shifts the location of dbd-A08a synapses, the authors conclude that axon guidance cues are the predominant signals that determine synaptic specificity in this circuit.

All reviewers are enthusiastic about the biological system, the question being addressed and the over-all quality of the data. It was generally agreed that this study will have broad appeal to the general audience of *eLife*. There were a number of issues that were commonly highlighted in the online discussion. These criticisms would primarily entail a more in-depth analysis of existing data and, therefore, it is generally believed that these comments can be addressed within the necessary timeframe for revision at *eLife*.

Essential revisions:

1) The authors use diffraction-limited light microscopy to measure "synapse voxels" – a proxy for synapse number that is not benchmarked. First and foremost, the authors should demonstrate that their synapse voxels correspond (or nearly correspond) to synapse numbers using an approach that provides the appropriate resolution to count synapses. For example, it might be possible to take advantage of Drep2-GFP or perhaps Dalpha7-GFP (acetylcholine receptor subunit; dbd is a cholinergic neuron).

2) A number of elements regarding light-level analyses should be described in greater depth. The authors state that the synapse voxels represent Brp puncta within 90 nm of A08a membrane. The Brp signal looks fairly diffuse in Figures 3 and 4. How were puncta versus background determined when setting the threshold for Brp puncta? Was the same threshold used across all images? Were Brp and membrane images thresholded independently, without looking at overlap during the threshold process? Since the Brp puncta images were dilated prior to looking for overlap with dendritic membrane, is the maximum distance 180nm (from one edge of the central pixel to the outer edge of the adjacent pixel)?

3) The authors should provide more information regarding the EM reconstruction and its use here i.e. how many neurons, the fraction of synapses observed between A08a/dbd close contacts. Of course the EM work is not high throughput, however from those neurons examined at the ultrastructural level it would be helpful to know not just that synapses were observed, but that they predominate when these particular pre- and postsynaptic partners are in close proximity.

4) The authors perform their imaging analyses to define the anatomy of the circuit in first instar larvae yet perform the functional imaging experiments in third instar larvae. Is the morphology still the same in third instars? The authors should provide evidence that the morphology and distribution of synapse voxels is similar at this later stage.

5) In the videos, it appears that ATR feeding increases baseline calcium activity in A08a, raising the possibility that functional connectivity could be altered in the presence of increased dbd activity throughout development. The authors should address why this difference in baseline calcium exists, if it holds across all videos. For example, are there any morphological changes to dbd or A08a under this rearing? If there exist data to address these ideas directly, this would be a nice addition. If not, further discussion should be added.

6) A related issue: some clarification of the optogenetic experiments appears warranted. Though "connectivity" as measured in this manner is shown not to be altered between the lateralized and wt larvae, are there data directly showing that in wild type selected ROIs from the lateral A08 dendrite are not activated, and in the lateralized dbd projections no activation of the A08 medial dendritic arbor region is observed?

7) Scale bars are inconsistently used. Please provide scale bars in Figures 2, 3, 4, and Figure 4—figure supplement 1 at all magnifications, not just the most highly magnified images.

---

## [Author Response]

Essential revisions:1) The authors use diffraction-limited light microscopy to measure "synapse voxels" – a proxy for synapse number that is not benchmarked. First and foremost, the authors should demonstrate that their synapse voxels correspond (or nearly correspond) to synapse numbers using an approach that provides the appropriate resolution to count synapses. For example, it might be possible to take advantage of Drep2-GFP or perhaps Dalpha7-GFP (acetylcholine receptor subunit; dbd is a cholinergic neuron).

As the reviewers described, we use a diffraction-limited light microscopy technique to quantify the position of putative synaptic contacts, or “synapse voxels”, between the dbd and A08a neurons. It is important to note that we are not claiming to count synapse numbers, but rather simply to measure synaptic material position along the mediolateral axis of the recipient A08a dendrite. We have clarified this in the text and figure legends, and thank the reviewer for highlighting this issue. In addition to measuring the *position* of synaptic material in wild-type and Unc-5 or Robo-2 misexpression larvae, we show that Chrimson activation of both wild-type and lateralized dbd can result in A08a calcium increases, suggesting the previously observed synapse voxels are relevant and functional.

We agree with the reviewer that it would be a step forward to express the Dalpha7::GFP postsynaptic marker in A08a to help distinguish dbd-A08a synapses from dbd synapses with other neurons, as the reviewer suggested. Unfortunately we were unable to add this component to our experiment, as we had no genetic space left to add this component (with five transgenes already present); there does not exist a lexAop-Dalpha7::GFP construct; and staining for GFP would also label dbd and A08a membranes, which already express a GFP variant, smGdP, that is recognized by GFP primary antibodies. And finally, dbd forms polyadic synapses (many post-synapses tightly grouped around a single pre-synaptic site) making it extremely difficult to say that a region of Brp+ voxels and Dalpha7::GFP voxels are the only synaptic site at that location. In the future, we hope someone publishes a paper on using Expansion Microscopy to link synapse number by light microscopy to synapse number by electron microscopy within the *Drosophila* ventral nerve cord – but this would deservedly be a standalone publication.

Overall, we feel confident that our “synapse voxels” are a good reporter for the *position* (not number) of putative dbd synapses along the A08a dendrite, and that our previous and newly added single photon and multiphoton optogenetic experiments show that both medial and lateral synapses are equally functional. And we thank the reviewer for helping us clarify the difference between synapse position and synapse number!

2) A number of elements regarding light-level analyses should be described in greater depth. The authors state that the synapse voxels represent Brp puncta within 90 nm of A08a membrane. The Brp signal looks fairly diffuse in Figures 3 and 4. How were puncta versus background determined when setting the threshold for Brp puncta?

The Brp signal was clearly discernable from background pixel intensity in all preparations analyzed, yet automatic thresholding algorithms (e.g. Huang or Otsu) available in ImageJ were unsuccessful in image segmentation, most likely due to the pixel intensity distribution. We found that manually assigning thresholds was the most accurate method of identifying Brp or membrane voxels.

Was the same threshold used across all images?

Because of the inherent variability in pixel intensity between different samples (due to the unavoidable variability of Gal4 and LexA expression), we could not assign the same threshold to all samples.

Were Brp and membrane images thresholded independently, without looking at overlap during the threshold process?

Yes, the Brp and membrane thresholds were assigned separately and *prior to* quantifying the number of overlapping voxels.

Since the Brp puncta images were dilated prior to looking for overlap with dendritic membrane, is the maximum distance 180nm (from one edge of the central pixel to the outer edge of the adjacent pixel)?

Our image analyses pipeline identifies the amount of Brp voxels that are either 1 voxel away or already overlapping. Since each voxel size is 90nm, then the “synapse voxels” represent the amount of voxels that have Brp less than 90nm away from membrane voxels.

We have added the above text to the Materials and methods.

3) The authors should provide more information regarding the EM reconstruction and its use here i.e. how many neurons, the fraction of synapses observed between A08a/dbd close contacts. Of course the EM work is not high throughput, however from those neurons examined at the ultrastructural level it would be helpful to know not just that synapses were observed, but that they predominate when these particular pre- and postsynaptic partners are in close proximity.

Thank you for the opportunity to elaborate on the EM data; in response we have added a new supplementary figure and a revised Table 1. We further analyzed the EM dataset to better understand if dbd and A08a always form synapses when these synaptic partners are in close proximity. Previous studies have implemented a “filling fraction” analysis to quantify the ratio of actual synapses to potential synapses (Stepanyants et al., 2002 and Gerhard et al., 2017). In the new Figure 2—figure supplement 1, we show that when the A08a membrane ‘skeleton’ is within a 2um distance from a dbd presynaptic site, then there is ~30% chance of synapse formation (its “filling fraction”). This is in the range seen for other *Drosophila* neurons (26-42%; Gerhard et al., 2017) or mammalian neurons (12-34%; Stepanyants et al., 2002). These data show that the dbd-A08a synapses in the EM database do not simply reflect proximity between the two neurons, but that dbd and A08a are indeed synaptically coupled at probabilities consistent with other neuronal types.

We have also added a revised Table 1 describing all neurons analyzed (A08a, and its input neurons dbd, A02l, A02d, A31x). We quantify the total number of synapses dbd, A02l, A02d, A31x input neurons make, the number they make on A08a, and the location of the synapses on A08a medial and lateral arbors. We show data from two hemisegments, illustrating the reproducibility of the connectivity. All neurons in Table 1 are fully annotated and there are no remaining unidentified synapses.

4) The authors perform their imaging analyses to define the anatomy of the circuit in first instar larvae yet perform the functional imaging experiments in third instar larvae. Is the morphology still the same in third instars? The authors should provide evidence that the morphology and distribution of synapse voxels is similar at this later stage.

Thank you for this important comment, it led us to add two new figures to the paper! We now show that dbd and A08a have the similar morphology at 72h ALH (the stage we did the functional experiments). We show that A08a still has a lateral, medial, and output domain; dbd still targets the medial domain and ignores the lateral dendrite and output domain. We have added this as a new Figure 5—figure supplement 1.

Importantly, we have also added two-photon optogenetic experiments which show that stimulation of dbd neurons will elicit an increase in GCaMP in A08a neurons, but stimulating the ventral off-target neurons generates little or no response in A08a. We are now extremely confident that dbd activity directly activates the A08a neurons, consistent with the direct synaptic connectivity seen in by EM.

5) In the videos, it appears that ATR feeding increases baseline calcium activity in A08a, raising the possibility that functional connectivity could be altered in the presence of increased dbd activity throughout development. The authors should address why this difference in baseline calcium exists, if it holds across all videos. For example, are there any morphological changes to dbd or A08a under this rearing? If there exist data to address these ideas directly, this would be a nice addition. If not, further discussion should be added.

That is a perceptive and accurate comment. Average baseline fluorescence for each individual experimental group was calculated by taking the average fluorescence in the three frames preceding the stimulus event. It turns out that the differences in baseline fluorescence between +ATR and -ATR groups are indeed statistically significant. Because the data is non-normally distributed, we used a Pairwise Wilcox Test followed by a Benjamini and Hochberg correction for multiple testing. The statistics support the idea that the animals fed ATR have a higher baseline level of A08a GCaMP fluorescence. One explanation for why +ATR animals have a higher baseline fluorescence could be that in between 561nm stimulus pulses we are still sampling with 488nm light (see Optogenetics Methods). It has been reported that Chrimson can be activated by these lower wavelengths (Klapoetke et al., 2014), so it follows that the baseline fluorescence would be higher for animals fed ATR. Importantly, this does not change our interpretation that lateralized dbd neurons form functional synapses with the A08a lateral dendrite. We have added the relevant text to the Discussion.

6) A related issue: some clarification of the optogenetic experiments appears warranted. Though "connectivity" as measured in this manner is shown not to be altered between the lateralized and wt larvae, are there data directly showing that in wild type selected ROIs from the lateral A08 dendrite are not activated, and in the lateralized dbd projections no activation of the A08 medial dendritic arbor region is observed?

We appreciate this comment and in response we have added a new figure panel and clarified the text. Low signal to noise made it difficult to measure GCaMP activity in the fine dendritic processes, whereas the signal was robust in the output region of the neuron where there are many pre-synaptic sites. We have seen the same thing when assaying GCaMP signal in motor neurons: it is very difficult to detect signal in the fine dendritic processes of single neurons, compared to their axonal domain. We have added a new panel (Figure 5C) and similar images in new Figure 6 to show the ROI used for quantifying GCaMP responses.

7) Scale bars are inconsistently used. Please provide scale bars in Figures 2, 3, 4, and Figure 4—figure supplement 1 at all magnifications, not just the most highly magnified images.

Scale bars were included in all single focal planes and standard maximum intensity projections. When the images were “3D projected” using Imaris software, we could not place a single scale bar because the foreground image is larger than the more distant part of the image, similar to a landscape painting where foreground trees are bigger than distant trees. We mention this in the methods. If the editor wishes, we can add scale bars and cite the region of the image for which they apply.